# Towards Kinase Inhibitor Therapies for Fragile X Syndrome: Tweaking Twists in the Autism Spectrum Kinase Signaling Network

**DOI:** 10.3390/cells11081325

**Published:** 2022-04-13

**Authors:** Claudio D’Incal, Jitse Broos, Thierry Torfs, R. Frank Kooy, Wim Vanden Berghe

**Affiliations:** 1Protein Chemistry, Proteomics and Epigenetic Signaling (PPES), Department of Biomedical Sciences, University of Antwerp, 2000 Antwerp, Belgium; claudio.dincal@uantwerpen.be (C.D.); jitse.broos@student.uantwerpen.be (J.B.); thierry.torfs@student.uantwerpen.be (T.T.); 2Department of Medical Genetics, University of Antwerp, 2000 Antwerp, Belgium; frank.kooy@uantwerpen.be

**Keywords:** fragile X syndrome, autism, intellectual disability, phosphorylation, protein kinases, phosphoproteomics

## Abstract

Absence of the Fragile X Messenger Ribonucleoprotein 1 (FMRP) causes autism spectrum disorders and intellectual disability, commonly referred to as the Fragile X syndrome. FMRP is a negative regulator of protein translation and is essential for neuronal development and synapse formation. FMRP is a target for several post-translational modifications (PTMs) such as phosphorylation and methylation, which tightly regulate its cellular functions. Studies have indicated the involvement of FMRP in a multitude of cellular pathways, and an absence of FMRP was shown to affect several neurotransmitter receptors, for example, the GABA receptor and intracellular signaling molecules such as Akt, ERK, mTOR, and GSK3. Interestingly, many of these molecules function as protein kinases or phosphatases and thus are potentially amendable by pharmacological treatment. Several treatments acting on these kinase-phosphatase systems have been shown to be successful in preclinical models; however, they have failed to convincingly show any improvements in clinical trials. In this review, we highlight the different protein kinase and phosphatase studies that have been performed in the Fragile X syndrome. In our opinion, some of the paradoxical study conclusions are potentially due to the lack of insight into integrative kinase signaling networks in the disease. Quantitative proteome analyses have been performed in several models for the FXS to determine global molecular processes in FXS. However, only one phosphoproteomics study has been carried out in *Fmr1* knock-out mouse embryonic fibroblasts, and it showed dysfunctional protein kinase and phosphatase signaling hubs in the brain. This suggests that the further use of phosphoproteomics approaches in Fragile X syndrome holds promise for identifying novel targets for kinase inhibitor therapies.

## 1. Introduction

Fragile X Syndrome (FXS) is the most prevalent form of inherited intellectual disability (ID) and autism spectrum disorders (ASD), and globally affects approximately 1:4000 males and 1:8000 females [1,2]. Patients exhibit distinct physical features such as a long face, prominent jaws, and elongated ears, in combination with macroorchidisms [3]. Furthermore, FXS patients also display mild to severe cognitive impairments and behavioral problems, including attention deficit hyperactivity disorder (ADHD), anxiety, memory impairments, and autism spectrum disorder (ASD) [4]. Epileptic seizures are also observed in 20% of FXS patients [5]. At the molecular level, the FXS is caused by a trinucleotide CGG repeat expansion in the 5′ untranslated region of the *Fragile X Messenger Ribonucleoprotein 1* gene (*FMR1*) [6]. This expansion is associated with loss of methylation boundaries, which are hypothesized to inhibit the spread of methylation of the *FMR1* promotor [7]. Loss of these insulator elements could contribute to total methylation of the promotor region and heterochromatin formation [8]. The spread of hypermethylation causes transcriptional silencing and an absence of the Fragile X Mental Retardation Protein (FMRP) [9].

FMRP is an RNA-binding protein that binds to polyribosomes supporting local protein synthesis [10]. Generally, FMRP represses the expression of specific mRNAs encoding pre- and postsynaptic proteins such as the N-methyl-D-aspartate receptor (NMDAR), metabotropic glutamate receptor 5 (mGluR5), postsynaptic density protein 95 (PSD-95), Homer1, PI3-Kinase Enhancer (PIKE), and voltage gated ion channels [11]. Loss of FMRP results in an increased protein translation of ~15–20%, heavily impacting various signal transduction pathways [1,12]. Indeed, hyperstimulation of mGluR5 as a result of FMRP silencing has been linked to aberrant translation of several neuronal proteins, and has been associated with dendritic spine abnormalities, causing mGluR-dependent long-term depression (LTD), increased seizure susceptibility, and accelerated prepubescent growth [13]. Genetic and pharmacological reduction of mGluRs corrects the deregulation caused by the absence of FMRP [2]. FMRP-related defects were also discovered in several neurotransmitter receptors pathways, of which the gamma aminobutyric acid receptor (GABA-R) is widely discussed. GABA is the main inhibitory neurotransmitter in the adult mammalian brain [14]. It is strongly involved in the modulation of neuronal activity. FMRP shows affinity for mRNAs encoding different GABA-R subunits. Hence, deregulation of the GABAergic system is thought to play a pivotal role in behavioral abnormalities underlying the FXS [15]. Another example of an FRMP-dysregulated protein is matrix metalloproteinase 9 (MMP9). MMP9-mRNA was found to be part of the FMRP complex, which clusters in dendrites. Loss of FMRP results in elevated synaptic MMP9 levels in *Fmr1* knock-out mice which, in turn, could contribute to impaired dendritic spine morphology and altered neuronal signaling [16]. MMP9 also cooperates with intracellular signaling molecules such as glycogen synthase kinase-3β (GSK3β), which is strongly activated in the *Fmr1* knock-out mice hippocampus. Aberrant GSK3β signaling has also been linked to structural changes in dendritic spines [17]. Another relationship was found between FMRP and the amyloid precursor protein (APP), linking the pathophysiology of the FXS with Alzheimer’s disease. Recently, it was shown that APP mRNA is repressed by FMRP [18]. Cleaving of APP by β-secretase results in the neurotoxic amyloid-β protein, of which the concentration is higher in FXS patients, possibly due to elevated APP synthesis [19]. Besides APP, another 4% of the total mRNAs in the brain show affinity for FMRP, including transcripts that regulate the synaptic cytoskeleton, such as activity-regulated cytoskeleton-associated protein (Arc), microtubule-associated protein 1B (Map1B), postsynaptic density protein 95 (PSD-95), and the ras-related C3 botulinum toxin substrate 1 (Rac1) [20]. These findings illustrate the involvement of FMRP in a multitude of intracellular pathways [1,21].

Most of the intracellular signaling pathways pivotal in the FXS are tightly regulated by protein kinases, which phosphorylate their protein targets [22]. Phosphorylation is a reversible post-translational modification (PTM) characterized by covalent addition of a phosphate moiety to an amino acid of a protein substrate [23]. This addition modifies the polarity of the protein, resulting in a more hydrophilic and polar character [24]. The change in polarity results in an altered conformation, inducing the formation of different protein–protein interactions or detachment from other complexes [25]. However, phosphorylation events are also strongly involved in the regulation of the biochemical activity of proteins, as well as the subcellular localization, degradation, and stabilization [26]. Any interference in the phosphorylation state will drastically affect their cellular function, and could potentially cause disease [27]. A much-debated example of a protein kinase in the FXS is mechanistic target of rapamycin (mTOR), which is involved in a translational pathway that is dysregulated by the loss of FMRP [28]. Phosphorylation of mTOR itself was reported to be increased in the hippocampus of juvenile *Fmr1* knock-out mice, enhancing phosphorylation of multiple downstream targets such as ribosomal protein S6 kinase beta-1 (S6K), 4E-binding protein (4E-BP), and eukaryotic initiation factor complex 4F (eIF-4E) [29]. In addition to this cascade of phosphorylation events following the loss of FMRP, many other phosphorylation/kinase abnormalities have been reported in the FXS [30]. In this review, we summarize the protein kinase and protein phosphorylation deregulation in the FXS. Our review aims to summarize the different studies reported and also to discuss potential novel therapeutic targets.

## 2. The Fragile X Syndrome: A Systemic Overview of the Molecular Pathophysiology

The FXS is an X-linked monogenetic disorder that is caused by a CGG trinucleotide repeat expansion in the 5′ untranslated region (UTR) of the *FMR1* gene [31]. The gene is located on Xq27.3 and is approximately 38 kb long [32]. During evolution, *FMR1* shows a high conservation in both rodents and primates [33]. Expression of the *FMR1* gene starts early during embryonic development, with the highest levels in brain, testes, ovaries, and thymus, concomitant with several symptoms underlying the FXS [34]. The *FMR1* gene has 17 exons which are alternatively spliced into 12 transcriptional variants [35]. Within the 5′ UTR, a CGG trinucleotide repeat is located (Figure 1A). Expansion of this repeat lies at the molecular basis of the FXS and can be subdivided into four allelic classes based on the expansion size of the repeat: normal (6–40 repeats), intermediate (41–60 repeats), premutation carriers (61–200 repeats), and the full FXS mutation (>200 repeats) [36]. Normal and intermediate alleles are considered to be transmitted in a stable manner without affecting the genotype [37]. In contrast, premutations are unstable and expand when passed on to the offspring. The ultimate full mutation is hypothesized to occur in utero, even before the stage of the zygote [38]. Premutations are characterized by a poor methylation status [7]. These alleles are transcribed and represent high *FMR1* mRNA levels. However, FMRP expression is slightly reduced [39]. The full FXS mutation corresponds with epigenetic modifications such as CGG repeat methylation, associated with transcriptional silencing of *FMR1*. Here, methylation of CpG-specific cytosines covers the entire promotor region, leading to heterochromatin formation [40].

The full repeat expansion and subsequent hypermethylation of the *FMR1* gene result in transcriptional downregulation and an absence of FMRP [41]. Mammalian FMRP is a 71 kDa protein that comprises a variety of functional domains (Figure 1B), including three RNA interacting domains of which two are K homology domains (KH1 and KH2), and an enriched cluster of arginine–glycine–glycine (RGG box). A nuclear localization signal (NLS) and nuclear export signal (NES) are also present within the amino acid sequence, enabling nucleocytoplasmatic shuttling [42]. A final feature is the presence of two tandem Agenet (Age) domains which have affinity for trimethylated lysine residues and could possibly interact with methylated histone H3K9 [43].

FMRP plays a crucial role as a negative regulator of translation of proteins that are involved in synaptic function, connectivity, plasticity, and dendritic morphology [44]. The way in which FMRP modulates translational repression is supported by several theories. First, monomeric FMRP is thought to dimerize in the cytoplasm, subsequently entering the nucleus through its NLS. Here, the dimer interacts with target-specific mRNAs through its KH-domains and RGG box. The FMRP–mRNA complex then leaves the nucleus by the NES. A first theory supports the concept of ribosome blocking. Subsequently, the FMRP–mRNA complex binds to the inter-ribosomal space, interfering with the binding of translation-specific initiation factors [41]. An alternative hypothesis suggests the involvement of the RNA-induced silencing complex (RISC). Here, FMRP represses the translation of synaptic mRNAs via miRNA interference. Only recently, FMRP was identified as a reader of N^6^-methyladenosine (m^6^A), a modification regulating mRNA function. Reading of the m^6^A modification enables shuttling of methylated mRNA targets between the nucleus and the cytoplasm during neural differentiation. Moreover, FMRP also interacts with another m^6^A reader YTH N^6^-methyladenosine RNA Binding Protein 2 (YTHDF2), which ensures degradation of the mRNAs previously stabilized by FMRP [45,46]. In the end, translationally silenced mRNAs are transported to the postsynaptic dendritic membrane, waiting for a translational activation signal [47]. The translation is, for example, initiated when the metabotropic glutamate receptor 5 (mGluR5) is activated upon ligand binding (Figure 2) [48].

In response to ligand binding, several postsynaptic membrane receptors can be triggered, initiating an intracellular signaling cascade. Common receptors still under debate regarding their involvement in the FXS are the metabotropic glutamate receptors (mGluRs), AMPA receptors, GABA receptors type A and B, NMDA receptors, and TrkB receptors (Figure 3). Signal transmission of one of these receptors will cancel out the translation inhibitory effect of FMRP, establishing protein synthesis. Activation of these membrane receptors results in an intracellular signaling cascade. A first pathway showing higher activation is the phosphatidylinositol 3-kinase (PI3K) pathway, resulting in upregulation of Akt and mTOR signaling. Simultaneously, ERK and TrkB signaling have been shown to be highly active with converging effects towards protein synthesis. In the case of the FXS, the inhibitory effect of FMRP is lost, supporting local protein synthesis of proteins such as Arc, Map1B, CamKII, postsynaptic density-95 (PSD-95), matrix metalloproteinase 9 (MMP9), and glycogen synthase kinase 3 beta (GSK3β). Elevated Arc levels will cause increased AMPA receptor internalization, resulting in ion channel imbalance. This causes a receptor imbalance, resulting in enhanced mGluR long-term depression (mGLuR-LTD), which contributes to altered hippocampal synaptic plasticity. Furthermore, the upregulation of key regulatory proteins such as Arc, Map1B, and PSD-95 causes deregulation of the neuronal cytoskeleton [15,35].

In addition to the glutamatergic imbalance, the GABA receptor system is also impaired in FXS patients. GABA is the principal inhibitory neurotransmitter expressed in the adult mammalian brain and is strongly involved in the modulation of neuronal activity. Dysregulated GABAergic transmission is thought to play a role in behavioral abnormalities underlying the FXS. Epileptic seizures and sleeping disorders are common problems affecting FXS patients. At a cellular level, GABA can bind to two different types of receptors, being the ionotropic GABA_A_ receptor and the metabotropic GABA_B_ receptor (Figure 3). Here, FMRP shows affinity for the mRNAs coding for the GABA_A_ receptor subunits α5, β2, and δ. Moreover, regulation of the mRNA subunit stability as well as inhibition of their degradation are FMRP-mediated effects. FMRP also shows affinity for the enzyme responsible for GABA synthesis, glutamic acid decarboxylase (GAD). Additionally, impairments of the GABA_B_ receptor are also reported. Upon binding of GABA, presynaptic vesicles, carrying glutamate neurotransmitters, are inhibited from unloading their cargo, resulting in an inhibition of the postsynaptic mGluR signaling. In case of the FXS, there is an imbalance of neurotransmission, favoring excitatory mGluR5 signaling [2,12,15].

## 3. The Fragile X Messenger Ribonucleoprotein (FMRP) Is an RNA-Binding Molecule, Which Is Tightly Regulated by the Actions of Protein Kinases and Phosphatases

FMRP is a multifunctional protein that plays a role in translational repression through ribosomal stalling, where it is locally modulated by posttranslational modifications such as phosphorylation and methylation (Figure 3). Initially, mass spectrometry analyses of murine brains and cultured cells showed that FMRP is mainly phosphorylated between the amino acid residues 483 and 521. Within this sequence, primary phosphorylation takes place at the conserved serine 499, which subsequently triggers phosphorylation of the nearby residues Ser489, Ser493, Ser496, Ser503, Ser510, and Ser513 [49,50]. Strict regulation of FMRP phosphorylation is crucial for controlling its ability to modulate the translation process. It is well established that signaling through neuronal GPCRs, including the metabotropic glutamate receptor (mGluR), impacts the phosphorylation status of neuronal proteins. Therefore, the translational activity of FMRP is regulated by the actions of the ribosomal protein S6 kinase 1 (S6K1) and protein phosphatase 2A (PP2A) system [6,51]. After mGluR stimulation, S6K1 is activated, and PP2A activity is inhibited, allowing phosphorylation of FMRP and its subsequent translational repression via ribosomal stalling [52]. In contrast to S6K1, PP2A initiates rapid FMRP dephosphorylation after group I mGluR stimulation, measured by enhanced PP2A activity. Furthermore, measurements up to five minutes after mGluR stimulation have shown mTOR-mediated PP2A inhibition, together with new phosphorylation events, indicating an activity-dependent induction of FMRP phosphorylation [53]. However, findings by Bartley et al. (2014) showed that FMRP phosphorylation at Ser 499 remained unchanged despite a higher mTORC1-S6K1 activity, thereby suggesting that other kinases may also play a role in FMRP phosphorylation [54,55]. Hence, casein kinase II (CKII) was discovered to phosphorylate FMRP at Ser499 within 2–4 h of synthesis, promoting dynamic phosphorylation of nearby residues by other kinase-phosphatase systems, also including S6K1/PP2A. Once FMRP is phosphorylated, it remains phosphorylated without affecting the half-life of the protein [50]. FMRP is also involved in activity-dependent protein translation which requires transport of mRNA into ribonucleoprotein organelles, called neuronal granules, which is facilitated by its C-terminal low-complexity disordered region. Here, post-translational modifications such as phosphorylation and methylation of FMRP show opposing activities after translation initiation. Both protein phosphorylation, as well as methylation, can modulate the ability of FMRP to bind mRNAs. For example, receptor stimulation promotes neuronal granule disassembly after FMRP dephosphorylation by PP2A. Here, methylation of FMRP causes a reduction in higher-order assembly formation with mRNA and polyribosomes, thereby activating protein translation. Reversely, PP2A activity is counteracted by the actions of S6K1, which induces FMRP phosphorylation, thereby promoting granule reassembly and translational silencing as well as FMRP demethylation [56]. These findings propose a model where FMRP is controlled by multiple phosphodynamic processes, all impacting the translational process [45,48,51].

## 4. Protein Kinases and Phosphatases Are the Main Drivers of Intracellular Signaling and Are Dysregulated in the FXS, Revealing New Strategies for Kinase Inhibitor Therapy

As mentioned above, the FXS is characterized by abnormal receptor signaling and intracellular second messengers, which are modulated by a bidirectional system, being protein kinases (PKs) and protein phosphatases (PPs) [57]. PKs can phosphorylate a complex amount of protein substrates by catalyzing the transfer of a phosphate group from ATP to specific amino acid residues [58]. These residues usually are serine, threonine, and tyrosine amino acids. Protein phosphorylation is the most common post-translational modification affecting the intracellular signal transduction and other cellular process such as neuronal growth, differentiation, learning, and memory [27]. In contrast to PKs functioning as primary effectors of phosphorylation, PPs act by removing the phosphate groups of phosphoproteins, a process called dephosphorylation [59]. Therefore, both phosphorylation and dephosphorylation dynamics are required for a balanced biochemical equilibrium [60]. Interference in the phosphorylation state can tremendously alter cell function, and could therefore contribute to disease [27]. For this reason, abnormalities in protein kinases and phosphorylation are thought to be involved in the molecular pathophysiology underlying the FXS [30,61]. In this section, we will review the most important protein kinases that are involved in the FXS (Table 1).

### 4.1. Serine-Threonine Protein Kinase B (Akt)

Serine-Threonine Protein kinase B (**Akt**) is a serine-threonine kinase, and its signaling is mainly activated by extracellular ligands such as growth factors [62]. In absence of these extracellular stimulants, Akt remains inactive, residing predominantly in the cytoplasm [63]. Binding of these factors to, e.g., receptor tyrosine kinases (RTK) or GPCRs will activate the phosphoinositide-3-kinase (PI3K) signaling pathway, generating the secondary messengers PIP_2_ and PIP_3_ [64,65]. These lipid messengers remove the inhibitory pleckstrin homology (PH) domain of Akt, anchoring the kinase into the membrane [66]. After membrane localization, Akt is activated by phosphorylation by several other kinases on residue Thr308 located in its activation loop, and Ser473 in the hydrophobic motif [67]. The activation loop is mainly phosphorylated by phosphoinositide-dependent kinase-1 (PDK1), while the phosphorylation of the hydrophobic domain is mediated by mTOR [68,69]. After these phosphorylation events, Akt detaches from the membrane and localizes to the nucleus, phosphorylating multiple downstream substrates, and thereby regulating various cellular processes such as apoptosis, cell cycle, metabolism, and gene transcription [70,71].

Akt signaling is central in FXS research and has been studied in the Fmr1 knock-out mouse model as well as in human specimens. Studies investigating the kinase activity of Akt report indistinct results, for example, levels of phosphorylated Akt were studied in the context of Ras signaling mechanisms that underly the impaired GluR1-dependent plasticity that is associated with FXS. Here, western blots showed no difference in the phosphorylation of Akt in cultured slices prepared from Fmr1 knock-out and wild-type mice, suggesting that Akt kinase activity is not affected in the FXS [72]. Another study investigated the PI3K/Akt signaling pathway and examined the phosphorylation of Akt at two activation sites, Ser473 and Thr308. They found that the phosphorylation of Ser473 was enhanced in the hippocampus of Fmr1 knock-out mice in comparison to wild-type mice. In line with this result, the phosphorylation of Thr308 was also shown to be increased in the Fmr1 knock-out mouse [29]. Next, Akt was studied in a mouse model of the FXS with treatment of lithium to modulate the cerebral protein synthesis. Upon assessment of the phosphorylation state of Akt in the hippocampus of control and lithium-treated mice, Liu et al. (2012) found that lithium decreased phospho-Akt at Ser473 in both wild-type and Fmr1 knock-out mice. Furthermore, they reported that the average phospho-Akt status was 16% higher in Fmr1 knock-out versus the wild-type mice [73]. In another pharmacological intervention to rescue GluA1-dependent synaptic plasticity and learning deficits in the Fmr1 knock-out mice model, a small upregulation in PI3K-Akt activity was observed [74]. Shidhu et al. (2014) provided evidence that matrix metalloproteinase (Mmp-9) is necessary in FXS development. Genetic disruption of Mmp-9 rescued dendritic spine abnormalities, mGluR5-dependent LTD, and FXS-related social behaviors. Furthermore, Mmp-9 deficiency reduced the phosphorylation levels of active Akt in the hippocampus of adult Fmr1 knock-out mice in comparison with wild-type mice [75]. Another study aimed to find a correlation in defects of the ERK and Akt pathways of platelets from FXS patients and Fmr1 knock-out neurons, and whether the drug lovastatin could influence these pathways. Western blot assessment of the phosphorylation of ERK and Akt before and after a 12-week lovastatin trial indicated that basal phosphorylation levels of ERK and Akt were increased in FXS platelets. Moreover, lovastatin specifically reduced the phosphorylation of ERK, but did not modify the Akt phosphorylation of Ser473 compared to baseline. Nevertheless, eight patients showed a slight increase in Akt phosphorylation [76]. However, Sawicka et al. (2016) did not find any evidence for a higher phosphorylation of Akt in the neocortex of 4-week-old Fmr1 knock-out mice. They also examined the activity of the PI3K/Akt/mTOR pathway by western assessment of Akt at Thr308, and reported that neither abundance nor phosphorylation of Akt was changed in the neocortex of Fmr1 knock-out mice relative to their wild-type littermates [77]. A last trial investigated whether carbamazepine could correct neuronal signaling, protein synthesis, and cognitive function in a Fmr1 knock-out mice model. Carbamazepine not only showed positive behavioral changes in Fmr1 knock-out mice, but also dampened the activity of the ERK and Akt signaling, as well as protein synthesis at the cellular level [78].

### 4.2. Adenosine Mono-Phosphate-Activated Protein Kinase (AMPK)

Adenosine mono-phosphate-activated protein kinase (**AMPK**) is serine-threonine protein kinase that consists of a heterotrimeric complex, comprising a catalytic α-subunit and two regulator β and γ-subunits [79]. The α-subunit accounts for the kinase activity domain with the critical residue Thr172. The β-subunit contains a carbohydrate-binding module (CBM), enabling AMPK to associate with effectors of glycogen metabolism [80]. The γ-subunit allows AMPK to adapt to changes in the cellular ATP to AMP ratio by its four tandem cystathionine-β synthase (CBS-domains) [81]. AMPK functions as a metabolic switch, regulating both anabolism by decreasing ATP consumption and catabolism by the stimulation of ATP synthesis [79].

AMPK signaling has also been shown to be affected in FXS and was identified as a new mechanistic link with insulin receptor signaling in a dFMR1 knock-out *Drosophila* fly model. The dysregulated insulin signaling was shown to affect circadian and memory deficits in the fly model [82]. In this study, knock-out flies were fed a Type 2 diabetic drug, Metformin, which increased PTEN expression and AMPK activity, while decreasing mTOR signaling. After treatment, dFMR1 flies improved significantly in short- and long-term memory. However, circadian rhythm deficits could not be rescued [82]. Gantois et al. (2017) took the Metformin treatment one step further and administered a daily dose for 10 days to Fmr1 knock-out mice. Treatment with Metformin did not persistently activate AMPK in the prefrontal cortex and hippocampus of Fmr1 knock-out mice upon western assessment of phosphorylated AMPK and its downstream targets Acetyl-CoA carboxylase 1 (ACC1), tuberin (TSC2), raptor, and B-Raf [83]. Besides the role that AMPK plays in regulating metabolic processes, it also has a significant role in autophagy. Autophagy was shown to be downregulated in hippocampal neurons of Fmr1 knock-out mice with enhanced mTOR activity, resulting in decreased phosphorylation of ULK-1 at Ser757. This enhanced mTOR signaling initiates an inhibitory signal, relieving the phosphorylation signal of AMPK at Ser 317. However, the activity of AMPK in autophagy signaling was not studied [84].

### 4.3. Calcium/Calmodulin-Dependent Protein Kinase Type II Subunit Alpha (CaMKIIα)

Calcium/calmodulin-dependent protein kinase type II subunit alpha (**CaMKIIα**) kinases consist of a family of effector enzymes that are responsive to calcium. They become active when intracellular calcium levels are elevated, subsequently phosphorylating various downstream protein substrates that are involved in synaptic vesicle formation, ion channel signaling, gene expression, and synaptic plasticity, memory, and learning [85,86,87]. Currently, four different CaMKII protein isoforms are known, namely α, β, δ, and γ [88]. CaMKIIs are composed of 12 subunits, forming a ring-link structure of six subunits [89]. The basic structure of each of the CaMKII subunits is highly conserved, independent of the protein isoform. Generally, the multimeric enzyme is composed of an amino catalytic domain, a regulatory domain with an auto-inhibitory site, and a binding domain for the Ca^2+^/calmodulin (CaM) complex, an isoform variable sequence, and ultimately an oligomerization domain at the carboxy terminus, allowing inter-subunit assembly [85,90].

Interestingly, the CaMKIIα isoform has been identified as an mRNA target of FMRP, and shows elevated protein levels in Fmr1 knock-out mice [91]. Guo et al. (2015) studied the dynamic regulation of the mGluR5–Homer system and identified a fast, activity-triggered dissociation of mGluR5 from the Homer scaffold protein which is regulated by phosphorylation by CaMKIIα. They also reported that CamKIIa mediates hyperphosphorylation of Homer at basal cellular measurements in cortical neurons of Fmr1 knock-out mice. Genetic or pharmacological inhibition, as well as treatment with dephosphomimetics, restored hyperexcitability and seizures in the Fmr1 knock-out mice [92].

### 4.4. Cyclin-Dependent Kinase 4 (CDK4)

Cyclin-dependent kinase 4 (**CDK4**) are a family of serine-threonine protein kinases that play a central role in cell cycle regulation. However, their function is not limited to mitotic cell division, as they also have crucial functions in other molecular processes such as migration, senescence, apoptosis, gene expression, and metabolism [93,94]. CDKs physically depend on a cyclin partner to become catalytically active, thereby phosphophorylating various proteins. Classification of different CDK family members was achieved by detection of a cyclin-box domain, responsible for binding and activation [95]. In context of the FXS, one member of the CDK family, in particular, stands out, being CDK4. In an investigation into whether Fmrp deficiency could affect adult neurogenesis, it was found that loss of FMRP increases the proliferation and changes the fate of adult neuronal progenitor cells. By co-immunoprecipitation with an FMRP-antibody, Cdk4 and Cyclin D were identified amongst other mRNAs such as Map1b, Eif1α, and Gsk3β, suggesting that FMRP regulates expression of proteins involved in adult neural progenitor cell function [96]. However, studies elucidating the activity of CDK4 rather than its expression have not yet been carried out in FXS research.

### 4.5. Diacylglycerol Kinase Kappa (DGKk)

Diacylglycerol kinases (**DGK**) are ubiquitously expressed enzymes, mediating a rate-limiting step in the diacylglycerol (DAG) signaling [97]. DAG is often synthesized after activation of GPCRs, controlling the cleaving PIP_3_ to IP_3_ and DAG mediated by phosphatidylinositol 4,5-bisphosphate by phospholipase C (PLC) [98]. The metabolization of DAG to phosphatidic acid (PA) is catalyzed by diacylglycerol kinase (DGK) [97]. The structure of DGK isoforms is highly diverse, whereby each isoform contains a specific subset of domains. Generally, each isoform contains a catalytic domain, empowering its kinase activity, and a C1-domain, enabling binding to DAG and DAG-like metabolites [99]. The DGK family comprises ten members, of which eight are detected in the mammalian brain, which indicates a pivotal role in development [100]. Moreover, each DKG isoform shows a region-specific expression pattern together with altered subcellular localizations in the central nervous system [101].

Among the CGK family members, the CGK kappa isoform has received the most interest. In an attempt to identify target mRNAs of FMRP, crosslinking immunoprecipitation experiments in cortical neurons demonstrated that CGKκ was a unique candidate. The loss of FMRP disrupted the CGKκ activity and expression, which is dependent upon group 1 metabotropic glutamate receptors. The decreased expression and activity of DGKκ correlated with dendritic spine abnormalities, synaptic plasticity aberrations, and behavioral abnormalities. Overexpression of DGKκ, to restore the reduced protein levels, rescued Fmr1 knock-out neurons from the dendritic spine defects, suggesting that DGKκ contributes to FXS pathology [102]. Similarly to the Fmr1 knock-out mouse brain, the expression of DGKκ was also decreased in post-mortem cerebellar extracts from FXS patients compared to unaffected controls [102]. In a follow-up study, DGKκ was administered to the brain of adolescent Fmr1 knock-out mice using adeno-associated viral vectors, thereby correcting the brain diacylglycerol and phosphatidic acid homeostasis. The in vivo rescue also restored FXS-related behaviors after a behavioral test battery assessment [103]. In another study, the DAG-signaling pathway was targeted with the Type 2 diabetes drug agonist, pioglitazone, to investigate its ability to correct the excessive lipid signaling and FXS-related behaviors in a Fmr1 knock-out model. Pioglitazone normalized the activity of protein kinase C (PKC) and the expression of eIF4E, a downstream effector of the DAG signaling pathway in cortical Fmr1 knock-out cultures. The agonist also caused an enhancement in the recognition of novel objects, normally absent in Fmr1 knock-out mice, suggesting that this drug affected long-term memory. Moreover, knock-out animals improved in social interaction, a key feature in autism-like behavior. Finally, a 19-day pioglitazone treatment also reduced testicular weight of Fmr1 knock-out mice, thereby improving macroorchidism, which is a key symptom of FXS patients [104].

### 4.6. Extracellular Signal-Regulated Kinases (ERKs)

Extracellular signal-regulated kinases (**ERKs**) enable cells to detect external stimuli such as growth factors, cytokines, stress factors, GPCRs, and intracellular calcium, allowing receptor-induced intracellular signaling [105]. The first discovered ERKs were identified as extracellular signal-regulated kinase 1 (ERK1) and extracellular signal-regulated kinase 2 (ERK2) [106]. Structurally, ERK1/2 kinases are classified among the mitogen-activated protein kinases (MAPKs), which act by phosphorylation upon activation of upstream MAPK kinases (MAPKKs) and MAPKK kinases (MAPKKKs) [107]. In this pathway, ERK1/2 function as a nodule signal integrator and transmit the signal by phosphorylation of a series of downstream effectors [108]. ERKs play a role in various cellular processes such as cell proliferation, differentiation, migration, survival, cellular development, cognition, and memory formation, thereby contributing to the FXS pathophysiology and idiopathic ASD in general [109,110]. Structurally, ERKs are composed of an amino-terminal (N) lobe and a larger carboxy-terminal (C) lobe, interconnected by a hinge-like structure [111]. Moreover, the structure contains two additional stretches, being a 30 amino acid sequence located between alpha helices, folding back in the C lobe and a C-terminal extension, assembling with the N lobe [112]. Activation of human ERKs is mostly associated with phosphorylation of Thr185 and Tyr187, inducing a conformational change in several polar elements within the structure [108]. ERK kinases are located within various subcellular localizations. Predominantly, activation of ERKs is attributed to the plasma membrane [113]. However, to activate their nuclear targets, they need to be imported into the nucleus from the cytoplasm. Later, the nucleocytoplasmatic distribution was found to be dependent on the oligomer status, with ERKs providing nuclear activation to exist as monomers, and dimers to associate with cytoplasmatic substrates [114].

A lot of effort has been made to determine the impact of ERK signaling in FXS, thereby exploring different sample types, time points, and species. A first study, using the selective group I mGluR agonist dihydroxyphenylglycine (DHPG) to evoke mGluR-dependent long-term depression, indicated that synthesis and degradation of FMRP were required as a bifunctional regulatory system controlling synaptic plasticity. Upon rapid stimulation of mGluRs, translation of mRNAs bound to FMRP is initiated and is excessive in Fmr1 knock-out mice. Here, the expression of ERK, whose mRNA is not bound to FMRP, was found to not be significantly different in Fmr1 wild-type compared to knock-out mice. Next, the activation, hence phosphorylation, of ERK was assessed, which is necessary for mGluR-LTP. Increased phosphorylation of ERK was found in Fmr1 knock-out mice, compared to wild-type mice. As ERKs play a pivotal role in protein synthesis, depending on mGluRs activation, it was investigated whether inhibition of the ERK pathway could affect the LTP. However, it was found that inhibition of the ERK pathway could not reverse the enhanced mGluR-LTP in Fmr1 knock-out mice [115]. Another study investigated abnormalities in pain-processing regulated by mGluR1/5, which is associated with self-mutilation behavior in FXS patients. Here, DHPG or a vehicle (control) were administered into the spinal cord of Fmr1 wild-type and knock-out mice. The phosphorylation of ERK was assessed using western botting 15 min after injection. Wild-type mice showed an increase in lumbar phospho-ERK1/2 status compared to vehicle-injected wild-type mice. However, this trend was not observed in DHPG and vehicle-treated knock-out mice, concluding that DHPG did not have the ability to induce ERK phosphorylation. However, basal phosphorylation ERK1 was found to be increased in Fmr1 knock-out mice, normalizing for total ERK expression [116]. Next, ERK signaling was investigated in the context of learning disabilities, which are associated with AMPA receptor trafficking. Phosphorylation of ERK1/2 was found to be unaffected in the CA1 cells of Fmr1 knock-out mice, although a higher activity of Ras was reported combined with comprised PI3K-Akt signaling, in comparison to Fmr1 wild-type mice [72]. Another study investigated the group I mGluR after stimulation with DHPG and subsequently isolated cortical synaptoneurosomes (SNS) from Fmr1 wild-type and knock-out mice. When studying the MAPK pathway, ERK was shown to be dephosphorylated after mGluR1/5 stimulation in Fmr1 knock-out mice; however, ERK was phosphorylated in Fmr1 wild-type mice. These results suggest aberrant protein phosphatase signaling downstream to mGluR1/5 stimulation. The activity of protein phosphatase 2A (PP2A) was assessed, which was found to be overactivated upon mGluR5 stimulation, rapidly dephosphorylating ERK. Ultimately, this imbalance could be rescued by treatment with phosphatase inhibitors such as okadaic acid, which restored baseline ERK activity [117]. The group I mGluRs were also investigated in other studies that determined the activity of PI3K and ERK in FXS. First, the p110β catalytic subunit of PI3K was immunoprecipitated from wild-type and knock-out cortical SNS. The precipitated catalytic subunit was subsequently used to convert PI to PI3-P and quantified using radiolabeled ATP. Here, PI3K induced a threefold increase in phosphorylation in Fmr1-deficient SNS. Secondly, ERK1/2 were co-immunoprecipitated from Frm1 knock-out SNS and used to induce phosphorylation in ELK-1. However, no significant difference in phosphorylation of ELK-1 was observed in Fmr1 knock-out compared to wild-type SNS. These findings suggest that the ERK signaling pathway was not primarily dysregulated in SNS of Fmr1 knock-out mice [118]. Another study investigated ERK signaling by using an in vitro assay, reflecting the excessive protein synthesis downstream of the mGluR5 in the hippocampus of Fmr1 knock-out mice. Here, acute inhibition of mGluR5 or ERK1/2 led to a reduction in elevated protein levels, compared to wild-type levels. However, inhibition of mTOR did not result in a correction of the excessive protein synthesis. In contrast to previous studies, Osterweil et al. (2010) reported that the mGluR5-ERK1/2 pathway is not always overactive in the Fmr1 knock-out hippocampus, but in absence of FMRP, specific mRNAs are more susceptible to ERK1/2 activation. Furthermore, they found that this higher susceptibility contributes to audiogenic seizures in the Fmr1 knock-out mouse, stating the need to target the ERK1/2 pathway and signaling pathway converging to ERK1/2 to develop future therapies for the FXS [119]. Focusing solely on the translation machinery, Bhattacharya et al. (2012) genetically removed p70S6 kinase 1, a master regulator of translation initiation and elongation. Reduction of S6K1 was achieved by crossing Fmr1 KO mice and S6K1 knock-out mice, which resulted in no aberrant physiological defects, and reproduced in normal frequencies with viable offspring. When assessing the phosphorylation state of the molecular network in which S6K1 is involved, they observed that the ERK phosphorylation in the hippocampus of Fmr1 and Fmr1/S6K1 knock-out mice was increased. However, the elevated ERK phosphorylation could be rescued by genetic ablation of S6K1, suggesting a crucial role in phosphorylation and regulation of translation in FXS [120]. A study by Hoeffer et al. (2012) attempted to translate the dysregulated translational signaling observed in the Fmr1 knock-out mouse model to FXS patients. An expression analysis of mRNAs that code for proteins involved in translational control was performed with lymphocytes derived from FXS patients and matched control subjects. An increased phosphorylation of eIF4E was observed, indicating an upregulation of translation in FXS patients. Interestingly, eIF4E is regulated by the signaling actions of ERK1/2, which was shown to be more strongly phosphorylated at threonine 202/204 in the Fmr1 knock-out model. For this reason, they also assessed levels of ERK1/2 phosphorylation in FXS subjects, which showed a trend towards higher phosphorylation but was not statistically significant, suggesting that ERK signaling is not the primary signaling pathway that is affected in the FXS [121]. Another key symptom of the FXS is ADHD, which is currently studied in a phase III trial with the drug metadoxine ER (MDX) in children and adults. In the Fmr1 knock-out mouse, MDX treatment improved attention, memory, learning, hyperactivity, and sociability, consistent with findings in clinical trials of MDX in adults affected by ADHD. At the molecular level, the Ras-Mek-ERK and PI3K-Akt-mTOR signaling pathways were tested for biomarker assessment, because they play a role in altered gene expression in the FXS. Brain levels of phospho-ERK were measured as an indicator for the intracellular ERK activity. Here, ERK activity was significantly upregulated in Fmr1 knock-out mice compared to their wild-type littermate controls. Treatment with MDX reversed the levels of phospho-ERK to the levels of wild-type mice that were MDX- and vehicle-treated. The normalization of the hyperactive ERK signaling encourages the potential of MDX treatment in the FXS and other ADHD-related cognitive disorders [122]. ERK signaling was also studied in FXS subjects, treated with the hypocholesterolemic drug lovastatin, inhibiting the mevalonate pathway which subsequently decreases membrane recruitment of Ras, thereby reducing ERK signaling. First, the controversial overactivation of ERK signaling was assessed by western blotting in resting platelets of 46 FXS patients and 38 controls. Here, phosphorylation of ERK was 1.64-fold elevated in FXS patients. Treatment with lovastatin corrected the ERK hyperactivation in FXS blood platelets, providing supporting data that ERK signaling can be used as a biomarker in FXS clinical trials [76]. Sawicka et al. (2016) also measured ERK signaling in the neocortex of Fmr1 knock-out mice. In line with most research on ERK signaling, they found elevated phospho-ERK levels in the neocortex of Fmr1 knock-out mice. Moreover, an alternative hub was identified regulating ERK signaling, namely, via S6 kinase p90-ribosomal S6 kinase (RSK), which phosphorylates ERK1/2 uniquely at Thr185 and Tyr187. Molecular interference of ERK signaling corrected the elevated RSK and S6 kinase activity. Remarkably, the blood–brain permeable RSK inhibitor BI-D1870 prevented audiogenic seizures in Fmr1 knock-out mice, suggesting a potential drug candidate for FXS treatment [77]. Research performed by Ding et al. (2020) investigated whether the FDA-approved drug carbamazepine could correct cognitive deficits in the Fmr1 knock-out mouse model. They found restoration of hippocampal-dependent memory in Fmr1 knock-out mice after carbamazepine treatment. At the cellular level, the total expression of ERK1/2 was not affected. However, Fmr1 knock-out mice were reported to have higher phospho-ERK1/2 levels, reflecting the protein kinase activity. Here, carbamazepine treatment restored the levels of phospho-ERK1/2 to baseline in wild-type and knock-out neurons, with increased treatment sensitivity in the knock-out neurons. Together, these results highlight the need for the re-evaluation of carbazepine for treatment of the FXS [78].

### 4.7. Focal Adhesion Kinases (FAK)

Focal adhesion kinases (FAK) and related proline rich tyrosine kinase 2 (Pyk2) comprise a family of non-receptor tyrosine kinases characterized by high sequence identity, conserved phosphorylation sites, and similar domain structures [123]. FAKs reside in the cytoplasm and are expressed in ubiquitous tissues, exerting essential roles in embryonic development and in human diseases such as cancer [124]. Structural analysis reveals four domains within the protein: (1) an N-terminal band four-point-one, ezrin, radixin, and moesin common domain (FERM), autoinhibiting the kinase activity and binding other proteins linking the plasma membrane and the cytoskeleton; (2) a poly-proline sequencing domain, enabling binding of proteins that contain an SH3 domain; (3) a central tyrosine kinase domain, and (4) a C-terminal focal adhesion targeting (FAT) domain, playing a role in the localization of the protein and downstream signaling [125]. FAKs are activated after binding of integrins or growth factor stimulation, after which they lose their autoinhibition signal and become autophosphorylated on Tyr402 or Tyr397, thereby creating a binding site for Src which can phosphorylate additional amino acid residues essential for its full kinase activity [126].

The role of FAK was only reported to a minor extent in FXS. Research performed by Sidhu et al. (2014) showed that mRNA of matrix metalloproteinase-9 (MMP9) is locally regulated by FMRP. Hence, loss of FMRP resulted in higher MMP9 activity in the Fmr1 knock-out mouse [127]. To further evaluate the effect of MMP9 on the FXS, a double knock-out mouse model was created that is deficient for both Fmr1 and Mmp9. Among other things, the effect of Mmp9 deficiency was investigated in the context of FAK activation. The level of FAK phosphorylation was assessed, indicative for its activity. An increase of 30% in phosphorylation of Tyr576 was reported in the Fmr1 knock-out mouse compared to wild-types. The elevated phosphorylation of FAK was then again reduced to wild-type levels in the Fmr1/Mmp9 double knock-out mice. However, these results were not statistically significant due to high biological variation of the tested samples. However, 14 days of in vitro administration of 100 ng/mL active MMP9 in wild-type hippocampal cultures was found to induce intracellular FAK kinase signaling due to cleavage of some extracellular matrix-associated integrin receptors [75].

### 4.8. Glycogen Synthase Kinase 3 (GSK3)

Glycogen synthase kinase 3 (**GSK3**) is a key enzyme involved in the regulation of glycogen metabolism, by phosphorylation of glycogen synthase [128]. Paradoxically, glycogen synthase phosphorylation by GSK3 inhibits its activity, thereby lowering the intracellular glycogen levels [129]. GSK3 has two paralogues, GSK3α and GSK3β. Naturally, GSK3s are active in the absence of growth factors, insulin, or small peptides, being phosphorylated at Tyr279 for GSK3α and Tyr126 for GSK3β. In their active state, GSK3 recognizes substrates that have been primed with a phosphate group, facilitating the ability to bind in a specific pocket. Subsequently, GSK3s are able to phosphorylate a specific serine or theonine residue four amino acids downstream of the substrate. However, upon binding of an agonist, GSK3α becomes phosphorylated by Akt, S6K, or MAPKs at Ser21 and GSK3β at Ser9, allowing the N-terminal tail to cover the priming phosphate binding pocket and thereby blocking the accessibility of the substrate proteins [129,130]. The GSK3α and GSK3β isoforms share a sequence homology of 85%, and there is an even higher conservation of 97% in their catalytic domain. Both isoforms are expressed ubiquitously in different areas of the brain and play a crucial role in brain bioenergetics, synaptic plasticity, proliferation, and survival [131].

As indirect evidence suggests that FXS patients could benefit from lithium treatment, there has been a growing interest in GSK3 kinase regulation in FXS. First, baseline expression levels were measured for both GSKα/β, and each isoform was shown to be present at equal expression levels in the Fmr1 knock-out and wild-type mice. To access the activity of GSK3 in the FX mouse brain, phosphorylation of Ser21-GSKα and Ser9-GSKβ were measured by western blotting, as this phospho-signal is indicative of inhibition of their activity. Both paralogues were found in a lower phosphorylation state in the Fmr1 knock-out mouse cortex and striatum compared to the wild-types, indicating increased GSK3 activity. In particular, the GSKα isoform was also dephosphorylated in the Fmr1 knock-out hippocampus. Similarly, lithium was able to reverse the effects of hyperactivated GSK3 signaling in the FX mouse brain by promoting the inhibitory Ser9–GSK3β phosphorylation. Treatment did not result in altered GSKβ expression levels, which supports the potential of lithium for the treatment of the FXS. Furthermore, the effect of mGluR5 antagonist MPEP was assessed, as it has been reported to affect common signaling pathways in the Fmr1 knock-out brain. A significant increase in the phosphorylation of Ser21-GSKα and Ser9-GSKβ was reported in the Fmr1 knock-out mouse, and MPEP treatment did not alter the expression of GSK3, indicating that both lithium and MPEP are both promising candidates for ameliorating FXS symptoms [132,133,134]. A study performed by Choi et al. (2016) highlighted the clinical potential of lithium, PDE-4 inhibitor, and mGluR5 antagonist treatment in a mouse model for the FXS. First, the effect of chronic lithium treatment on GSK3β phosphorylation was investigated. Here, a slight increase in the phosphorylation of GSK3β was observed, when normalizing for total GSK3β levels. Secondly, the effect of lithium treatment was explored in Tau-mediated phosphorylation, as Tau phosphorylation at Ser202 is a known substrate of GSK3β. The basal expression levels of Tau were increased in the Fmr1 knock-out mouse compared to their wild-type littermates, and there was also a higher degree of phosphorylation to total Tau in Fmr1 knock-out mice versus wild-type mice. Lithium reduced the total expression of Tau in Fmr1 knock-out mice in comparison to a vehicle control but did not affect Tau expression in wild-type mice. However, chronic exposure to lithium significantly decreased the phosphorylation of Tau in comparison to vehicle controls, indicating that the activity of GSK3β is reduced after lithium treatment in Fmr1 knock-out mice. The effect of treatment with the group II mGluR antagonist LY341494 was studied in the context of GSK3β phosphorylation, as antagonizing mGluRs is known to elevate PKA levels, thereby lowering GSK3β activity. The effect of a vehicle control did not affect GSK3β expression, but LY341494 treatment did reduce global GSK3β levels in Fmr1 wild-type and knock-out mice. Phosphorylation of GSK3β normalized to total GSK3β was higher in Fmr1 knock-out vehicle-treated mice than wild-type vehicle-treated mice. LY341494 treatment did not affect GSK3β phosphorylation in both genotypes after vehicle treatment. Moreover, LY341494 treatment also significantly reduced phosphorylation of the GSK3β target Tau at Ser202 in Fmr1 wild-type compared to their vehicle controls. Lastly, the PDE4 inhibitor rolipram was examined. Here, vehicle treatment, as well as rolipram, increased total GSK3β protein levels. Rolipram treatment had no effect on the phospho/total GSK3β ratio, although there was a trend towards increased phosphorylation in the Fmr1 knock-out mouse. Similarly, chronic rolipram treatment decreased phospho-Tau levels in the Fmr1 knock-out mice compared to vehicle-treated Fmr1 knock-out mice. These results suggest that chronic treatment of rolipram targets GSK3 signaling and holds promise for future clinical testing [135]. Although a lot of research has been performed on GSK3 signaling in the FXS, the use of GSK3 inhibitors failed in clinical trials because of toxicity, as these inhibitors inactivate both GSK3 paralogues [136]. For this reason, McCamphill et al. (2021) designed GSK3 paralogue-specific inhibitors and investigated the potential of these compounds in vivo in the FX mouse model. They demonstrated that inhibition of GSK3α corrected the elevated protein synthesis, audiogenic seizures, and hyperexcitability in Fmr1 knock-out mice. Inhibition of GSK3β did not improve the FX phenotype. Inhibition of both paralogues prevented NMDA receptor-dependent LTD in the hippocampus. However, only inhibition of GSK3α rescued the LTD and also corrected the deficits in learning and memory in Fmr1 knock-out mice. These results indicate that selective inhibitors modulating GSK3α activity could provide therapeutic outcomes for treatment of the FXS [137].

### 4.9. LIM Domain Kinase 1 (LIMK1)

LIM Domain Kinase 1 (**LIMK1**) is a serine/threonine kinase that is involved in the regulation of the actin cytoskeleton, which is predominantly expressed in neuronal tissues such as the hippocampus, where it plays a critical role in memory and learning [138]. Structurally, LIMK1 comprises a C-terminal kinase domain together with two N-terminal LIM domains and one PZD domain [139,140]. LIMK1 can regulate its own kinase activity by binding of the LIM domains to the kinase domain [141]. In response to external signals, members of the Rho small GTPase family, such as RhoA, Rac1, and Cdc42, are activated, mediating actin dynamics though LIMK1 signaling [142,143]. Rho GTP-ases are able to regulate the activity of LIMKs by Rho kinases (ROCKs) and p21-activated kinases (PAKs), which can both phosphorylate Thr508 and Thr505 of LIMK1 and increase its activity [144,145]. Phosphorylation of Ser323/596 mediated by protein kinase A (PKA) also contributes to LIMK1 activity [146]. LIMK1 is also tightly regulated by slingshot protein phosphatase (SSH), which lowers LIMK1 activity by dephosphorylating Thr508 [147]. Upon activation, LIMK1 phosphorylates cofilin at Ser3, thereby preventing the binding of cofilin at actin filaments [148]. Moreover, key neuronal transcription factors, such as CREB and Nurr1, have also been identified as substrates of LIMK1 [149,150].

The research on LIMK1 has only recently been explored. In 2017, bone morphogenic protein type II receptor (BMPR2) mRNA was identified as a target of FMRP. Absence of FMRP, which is the case in the FXS, increased intracellular BMPR2 levels and activated LIMK1. Treating postnatal Fmr1 knock-out mice by pharmacological inhibition of LIMK1 corrected morphological abnormalities in the FX model. Moreover, BMPR2 and markers for LIMK1 activity were studied in post-mortem prefrontal tissue from FXS patients and were found to be elevated in FXS patients, suggesting that BMPR2-LIMK1 signaling is a possible target for future therapy of the FXS [151]. Later, in 2018, Rho GTPase signaling was investigated in the FXS mouse model. Here, a higher activity of Rac1 was found in the Fmr1 knock-out somatosensory cortex, which impairs the actin regulator cofilin. Although cofilin expression levels were not significantly altered in FX cortex, phosphorylation of cofilin at Ser3 was found to be increased in whole-cell lysates and synaptosomes of the somatosensory cortex of Fmr1 knock-out mice. The degree of actin polymerization was explored by determining the F-actin/G-actin ratio. Indeed, FX-mice showed a threefold increase in the F-actin/G-actin ratio in comparison with wild-type mice, suggesting that cofilin cannot depolymerize the actin filaments. The phosphorylation of cofilin is mediated by LIMK1 though Rac1 and effector PAK1 and was found to be increased at Thr508 in isolated synaptosomes in Fmr1 knock-out mice. Ultimately, the FX-mouse phenotype was rescued by viral delivery of an active cofilin mutant into the somatosensory cortex which rescued the immature dendritic spines. Altogether, Rac1-PAK1-LIMK1-cofilin signaling could be a promising new target for kinase inhibitor therapy [152].

### 4.10. MAPK Interacting Serine/Threonine Kinases (MNK)

The family of MAPK Interacting Serine/Threonine Kinases (**MNK**) is involved in regulation of mRNA translation and comprises two members, being MNK1 and MNK2 [153]. Upon activation by ERKs or MAPKs, MNKs finetune the translation process by phosphorylating the cap binding eukaryotic initiation factor 4E (eIF4E) [154]. Both members share a high sequence similarity and resemblance in structure [153]. The MNK members are activated upon different stimuli, e.g., MNK1 is activated by growth factors, radiation, mitogens, stress, and cytokines. However, MNK2 shows a rather high basal activity with more resistance to ERK inhibition compared to MNK1 [155]. Structurally, both MNK1/2 contain an N-terminal polybasic amino acid region, mediating their subcellular localization. Moreover, a catalytic domain is present with conserved MAPK phosphorylation sites together with a C-terminal MAPK binding domain of protein kinases such as ERKs and p38 [156,157]. MAPK stimulation of MNKs induces Thr197 and Thr202 phosphorylation in the catalytic domain, enhancing the binding to the eukaryotic initiation factor 4G (eIF4G). In parallel, MNK1 also phosphorylates eIF4E, releasing it from eIF4G to regulate mRNA translation [158]. In contrast, the activity of MNK can be inhibited by protein phosphatase 2A (PP2A) [159].

As loss of FMRP resulted in eIF4E hyperphosphorylation in the Fmr1 knock-out mouse model, MNKs activity was suggested to contribute to phenotypes associated with the FXS due to an increased MAPK signaling. First, genetic removal of MNK restored the core phenotypes described in the Fmr1-deficient mouse model. Here, one single substitution of an amino acid in eIF4E at Ser209A resulted in a decreased MNK-mediated phosphorylation [160]. Later, a highly specific inhibitor of MNK, tomivosertib (eFT508), was tested in the FX mouse model aiming for reduction of eIF4E phosphorylation. The inhibitor was capable of crossing the blood–brain barrier and reduced eIF4E phosphorylation of Ser209 in the cortex, striatum, and hippocampus of Fmr1 knock-out mice after intraperitoneal injections with a varying dose range of 1–8 mg/kg. Treatment with eFT508 restored hyperactive and anxiety-related behaviors, synaptic signaling, memory, and learning abnormalities, as well as aberrant social interactions. Together, these findings bring eFT508 to the forefront as a potential treatment for the FXS [161].

### 4.11. Mechanistic Target of Rapamycin Kinase (mTOR)

Mechanistic Target of Rapamycin Kinase (**mTOR**) is a serine-threonine kinase and acts as a master regulator of diverse biological processes such as ribosome biogenesis, mRNA translation, cell growth, and neuronal plasticity. The main function of mTOR is regulation of the translational machinery after activation of ligands, growth factors, or insulin, and protein kinases such as Akt and PI3K. It fine-tunes translation by activating p70 ribosomal S6 kinase (p70S6K) and inhibiting eIF4E binding protein (4E-BP1) [162]. The mTOR has a high conservation in its structure: at the N-terminal, it contains twenty tandem HEAT (Huntington, EF3, A subunit of PP2A, TOR1) repeats, mediating protein–protein associations. Moreover, a FAT (FRAP, ATM, TRAP) domain together with two lobes forming a kinase domain (N-KD and a C-KD) are present [163]. The mTOR exerts its function by associating with other proteins, forming two different complexes named mTOR complex 1 (mTORC1) and 2 (mTORC2) [164]. The mTORC1 is a sensor for several metabolites such as glucose, amino acids, and ATP, but it also senses growth factors and, to a certain extent, neurotransmitters. Upon activation, mTORC1 regulates protein synthesis, metabolism, and autophagy. On the other hand, mTORC2 shows activation exclusively after growth factor stimulation, thereby controlling survival, apoptosis, proliferation, and cell shape [165]. The mTORC1 and mTORC2 complexes share some of the proteins they associate with, such as DEPTOR and the Tti1/Tel2 complex, required for assembling and stability of the complex. However, both complexes also contain their unique members. The mTORC1 associates with two additional members: Raptor (regulatory associated protein of TOR), and activator PRAS40 (proline-rich Akt substrate of 40 kDa). The mTORC1 contains three additional subunits: Rictor (rapamycin-insensitive companion of mTOR), mSin1 (mammalian stress-activated Map kinase-interacting protein 1), and protein observed with Rictor 1 and 2 (Protor1/2) [166].

As mTOR controls several nervous system functions such as development, plasticity, memory, and cognition, it is no surprise that neurological disorders such as FXS report abnormal mTOR signaling [167]. An early report by Osterweil et al. (2010) showed that the Fmr1 knock-out mouse is typified by elevated protein synthesis, according to the mGluR5 theory. The elevated protein synthesis could be reduced to wild-type levels after inhibition of mGluR5 or ERK1/2. However, inhibition of mTOR by rapamycin could not reverse this biochemical phenotype. As a read-out for the mTOR activity, the phosphorylation of p70 S6 kinase (S6K), a direct downstream target, was assessed. Here, a significant decrease in phosphorylation was observed at Thr389 in Fmr1 knock-out and wild-type mice after rapamycin treatment, suggesting that mTOR signaling does not contribute to the elevated protein synthesis and perhaps regulates LTD by means of other compensatory mechanisms [119]. At the same time, findings by Sharma et al. (2010) were published, contrasting the results of Osterweil et al. (2010). They found that mTOR phosphorylation was increased at Ser2448 in whole hippocampal lysates or isolated postsynaptic densities of Fmr1 knock-out mice. Total mTOR expression did not differ in genotype. Next, four alternative experiments were performed to independently examine the kinase activity of mTOR. In the first experiment, the association of raptor with mTOR was studied under basal conditions and upon mGluR stimulation. Under basal conditions, the association was greater in knock-out mice than in wild-type mice, suggesting an enhanced activity. Stimulation with the mGluR1 agonist DHPG did not alter the raptor-mTOR association. In a second experiment, the kinase activity of mTOR was assessed. The mTOR was immunoprecipitated from the hippocampal CA1 area of knock-out and wild-type animals and was subsequently brought in proximity with purified S6K. Here, an increased kinase activity was observed. In a third experiment, the phosphorylation status of S6K was assessed by western blotting. Phosphorylation at Thr389 was significantly increased in hippocampal lysates from juvenile Fmr1 knock-out mice. In parallel, another direct target of mTOR, the eIF4F initiation complex, was assessed. Typically, 4E-BP binds to eIF4E, preventing association with eIF4G. Upon mTOR activation, 4E-BPs are phosphorylated, enabling eIF4G/eIF4E complex formation. In line with the expectations, 4E-BP was characterized by a higher degree of phosphorylation in the FX mice. Moreover, co-immunoprecipitation experiments demonstrated increased levels of eIF4G/eIF4E in Fmr1 knock-out mice, indicative of elevated mTOR activity. Consistent with these results, mGluR-LTD at CA1 synapses was stronger in Fmr1 knock-out mice and was not responsive to rapamycin treatment, suggesting mTOR deregulation in FXS, which interferes with mGluRs activity and synaptic plasticity [29]. Later, Bhattacharya et al. (2012) genetically removed S6K by creation of an S6k/Fmr1 double knock-out mouse. As S6K has the ability to phosphorylate mTORC1 at Ser2448, the mTOR activity was assessed in hippocampal lysates. Phosphorylation of mTOR was increased in the Fmr1 knock-out mouse, and genetic removal of S6K in the double knock-out mouse rescued the increased mTOR phosphorylation, thereby correcting the elevated protein synthesis. These findings underline that restoration of translation could restore some biochemical phenotypes of FXS [120]. Translational research of these findings was performed by Hoeffer et al. (2012), who investigated elevated protein synthesis in lymphocytes of FXS patients and healthy controls. Surprisingly, when assessing the phosphorylation of mTOR at Ser2448, no significant differences were found in phospho-mTOR to total mTOR expression in lymphocytes of FXS patients versus controls. However, phosphorylation of substrates and regulators of mTOR were found to be increased, such as S6K1 at Thr389, Akt at Ser473, eIF4E at Ser209, and ERK1/2 at Thr202/Tyr204. These results indicate that not mTOR itself, but the mTOR signaling pathway, is altered and contributes to elevated protein synthesis in FXS patient-derived lymphocytes [121]. Another study investigated the elevated protein synthesis in the cerebrum of FX mice, and whether administration of lithium could reverse these core biochemical parameters. They reported an increase in protein synthesis with a rate of 3–20% in Fmr1 knock-out mice compared to wild-type mice. Lithium treatment decreased protein synthesis in both genotypes, but the results were only significant in the Fmr1 knock-out mice model. Moreover, key signaling hubs were analyzed, such as the Akt-mTOR signaling pathway. Although no differences were found in the phosphorylation status of Akt, higher phosphorylation of mTOR at Ser2448 was found in untreated Fmr1 knock-out mice. Global expression of mTOR was unaffected in both genotypes. Lithium lowered the elevated mTOR phosphorylation to baseline levels in the FX mouse. However, an increase in phosphorylation was reported in wild-type mice after lithium administration, indicating genotype differences in lithium response. Besides western blotting, phospho-mTOR was also assessed using immunohistochemistry. High levels were reported in the dentate gyrus and pyramidal cell layers of CA1. There was also increased immunoreactivity in the pyramidal cells of untreated Fmr1 knock-out mice compared to wild-types. Lithium treatment reversed the enhanced phosphorylation in the Fmr1 knock-out mouse, suggesting lithium as a potential treatment candidate for the FXS in clinical trials [73]. In 2013, the endocannabinioid system was targeted in a mouse model for the FXS. This system was capable of enhancing the PI3K-Akt-mTOR-p70S6K pathway in the hippocampus, and regulates plasticity, cognition, anxiety, and seizures. Pharmacological inhibition and genetic ablation of one of the receptors of the endocannabinioid system (CB1R) resulted in a rescue of cognition, audiogenic seizures, and altered dendritic spines. In addition, the overactive mTOR signaling pathway was rescued, which was elevated in the FX mouse hippocampus. Blocking of another endocannabinioid receptor (CB2R) resulted in anxiety-like behavior, suggesting that the endocannabinioid system is a relevant target for FXS treatment [168]. Matrix metalloproteinase 9 (MMP9) was also identified as a target of FMRP and its translation is also regulated by FMRP. Sidhu et al. (2014) created an Fmr1/Mmp9 double knock-out mouse model and studied whether this genetic disruption could correct some FX-related phenotypes. In this new mouse model, phosphorylation of mTOR and its downstream effectors eIF4E and 4E-BP were assessed by western blotting. An increase of 50% in the phospho-mTOR/mTOR ratio was reported in the Fmr1 knock-out mouse. However, the elevated phosphorylation rescued by genetic ablation of Mmp9 in the double knock-out mouse and mTOR phosphorylation decreased to wild-type levels. The same findings were observed for the ratios of phosphorylated eIF4E/eIF4E and 4E-BP/4E-BP, indicative of disrupted mTOR signaling in the Fmr1 knock-out mouse model [75]. Research by Choi et al. (2016) investigated three different treatment strategies, being lithium, an PDE-4 inhibitor and an mGluR5 antagonist, in a mouse model for the FXS. First, the effect of chronic lithium treatment was investigated. The expression of mTOR was unaffected in Fmr1 knock-out mice compared to vehicle-treated wild-types. Lithium treatment did reduce the expression of mTOR in wild-type mice, but not in the FX model. The phosphorylated mTOR at Ser2448 relative to total mTOR expression was increased in Fmr1 knock-out mice versus wild-type mice. Chronic administration of lithium reduced the ratio of phosphorylated mTOR in Fmr1 knock-out mice versus vehicle-treated wild-types. However, no decrease in of phosphorylation was present in wild-type mice. These results indicate that chronic administration of lithium lowers mTOR signaling in a mouse model of the FXS. Next, the effect of treatment with the group II mGluR antagonist LY341494 was studied for mTOR phosphorylation. The expression of mTOR was unaffected in Fmr1 knock-out and wild-type mice after vehicle treatment or chronic LY341494 treatment. The ratio of phosphorylated mTOR to total mTOR was decreased in both genotypes after vehicle control, but not after chronic mGluR antagonist treatment. Lastly, the effect of the PDE4 inhibitor rolipram was examined. Here, no differences were detected in total mTOR expression between vehicle-treated wild-type and Fmr1 knock-out mice. However, rolipram treatment elevated the mTOR expression in both genotypes. The ratio phosphorylated to total mTOR remained unaffected pre- and post-rolipram treatment for both genotypes, indicating that only lithium had the potential to reduce the elevated mTOR activity in the Fmr1 knock-out mouse model [135]. Another study investigated mTOR signaling in the neocortex of 4-week-old Fmr1 knock-out mice, as this area was reported to have increased mRNA translation. In whole cell lysates, mTOR phosphorylation, indicative for its activity, was assessed, as well as regulators of mTOR, such as upstream kinases PI3K and Akt together with the downstream effector 4E-BP. There was no significant difference in expression nor phosphorylation of mTOR at Ser2448, Akt at Thr308, or 4E-BP at Thr37/46 in the neocortex of Fmr1 knock-out mice compared to their littermate controls. However, phosphorylation of another downstream target, ribosomal protein S6, was increased at Ser 235/236, despite normal mTOR activity, suggesting that the elevated protein synthesis reported in the FXS is not only related to mTOR signaling but could also be affected due to alternative or compensating signaling pathways [77]. In 2018, a study reported drastic negative effects of chronic rapamycin, an mTORC1 inhibitor, treatment on behavior in the Fmr1 knock-out mouse model. Here, mTORC1 activity was found to be increased in the Fmr1-deficient mouse model. Chronic treatment of rapamycin worsened behavioral effects such as passive avoidance and audiogenic seizures. Phosphorylation of S6 was found to be increased in the untreated Fmr1 knock-out mice, and rapamycin treatment normalized this elevated phosphorylation. However, chronic administration of rapamycin had dramatic effects regarding social behavior and sleep, impairing the Fmr1 knock-out mice. These results indicate that mTOR signaling is likely not an effective therapeutic target [169]. Only recently was a possible involvement of autophagy in ASD shown, which initiated a novel therapeutic area for research in the FXS. Yan et al. (2020) investigated the autophagic flux in hippocampal neurons of Fmr1 knock-out mice and reported a downregulation of autophagosome formation and autophagic flux. The underlying mechanisms were studied, in which mTOR phosphorylation was assessed, as mTOR is an activator of ULK-1, a component of the autophagy signaling pathway. Increased phosphorylation of mTOR was reported at S2448, indicative of a higher mTOR kinase activity in the Fmr1 knock-out neurons. Phosphorylation of ULK1 at Ser757 was also elevated in Fmr1-deficient neurons, suggesting a downregulation of autophagy as the Ser757 residue is a marker for antiautophagy. The crucial partner of mTORC1, being Raptor, was examined, because it facilitates S6K to the lysosomes, inhibiting autophagy by ULK-1 sequestering. Raptor expression was unaffected in whole-cell lysates of Fmr1 knock-out and wild-type neurons. However, after subcellular fractionation of hippocampal fragile X neurons, Raptor expression was increased in the lysosomal fraction, indicating a higher mTORC1 activity. Immunostaining of capsine D, a lysosomal marker, also colocalized with Raptor and was increased in hippocampal Fmr1-deficient versus wild-type neurons. Together, these results indicate a downregulation of autophagy in Fmr1-deficient neurons, and a higher localization of mTORC1 at lysosomes, implicating a deregulation of mTOR kinase activity in the FXS [84].

### 4.12. p21 (RAC1) Activated Kinases (PAKs)

The family of p21 (**RAC1**) Activated Kinases (PAKs) consist of a diverse group of serine and threonine kinases that act as main effectors of Rho GTPases Cdc42 and Rac. They function as a molecular switch, regulating the cytoskeleton and neuronal activity [170,171]. To date, six mammalian PAK isoforms have been discovered, and they are categorized into two groups based on their structure and sequence homologies. A first group comprises PAK1, PAK2, and PAK3, whereas a second group consists of the PAK4, PAK5, and PAK6 isoforms [172]. Although group I shares higher sequence similarity than group II PAKs, they overall contain the same protein motifs with some unique differences. Globally, they contain a p21-binding domain (PBD) and a serine/threonine kinase domain in the C-terminus, together with an autoinhibitory domain (AID) at the N-terminus [173]. The PBD domain functions as a scaffolding site for binding of Cdc42 and Rac GTPases, while the AID domain interacts with the kinase domain of another PAK, thereby assembling an inactive homodimer [174,175]. Ultimately, the homodimer disassembles after binding of active Rho GTPases to the PBD domain, thereby autophosphorylating multiple internal sites in a monomeric confirmation for recruitment of adaptor proteins such as NCK adaptor protein 1 (NCK1), growth factor receptor-bound protein 2 (GRB2), and nucleotide-exchange factor PIX [175,176].

As a higher spine density and elongation, together with cytoskeletal abnormalities, are frequently reported in FXS, and loss-of-function mutations in the PAK3 gene are known to result in X-linked intellectual disability, it is no surprise that PAK signaling could be involved in its molecular pathophysiology [177,178]. A study by Hayashi et al. (2007) created a double knock-out mouse by crossing a dominant negative PAK transgenic mouse with an Fmr1 knock-out mouse. PAK activity was not assessed in the double-knock-out mouse; however, it was assumed that the activity would start to reduce at the age of three weeks, as these findings were reported in the dominant negative PAK transgenic mouse model after measuring the levels of active, autophosphorylated PAK. In the dnPAK/Fmr1 double knock-out mouse model, the interactions of PAK1 and FMRP were studied. Immunoprecipitation with a PAK1 antibody showed reactivity after immunolabeling with an FMRP antibody, indicative of a direct interaction of both proteins. Furthermore, in vitro mutants of FMRP at crucial sites such as the RGG box and the phosphorylation domain Ser499 were still capable of binding PAK1. However, mutating FMRP at the KH domains resulted in an inability to bind PAK1, suggesting that PAK1 binds FMRP by its KH domains. A genetic reduction of PAK also rescued cortical LTP and partially restored locomotor activity, stereotypy, anxiety, and trace fear conditioning. These results indicate that genetic reduction of PAK is a potential target for FXS therapy [179]. Later, Dolan et al. (2012) treated the Fmr1 knock-out mouse model with a potent small molecule PAK inhibitor FRAX486. This compound was tested to exclusively inhibit group I PAKs and was able to penetrate the blood–brain barrier. Treatment resulted in rescue of the abnormal dendritic spine morphology and reduced audiogenic seizures in the Fmr1 knock-out mouse. In addition, hyperactivity and repetitive behaviors were ameliorated. These two studies provide evidence that genetic reduction of PAK, as well as pharmacological inhibition of PAK, can restore spine abnormalities and behavioral symptoms relevant to the FXS [180].

### 4.13. Phosphatidylinositol-4,5-Bisphosphate 3-Kinases (PI3Ks)

Phosphatidylinositol-4,5-Bisphosphate 3-Kinases (**PI3Ks**) comprise a group of plasma membrane-associated lipid kinases, which phosphorylate a hydroxyl group on position three of the ring structure of inositol glycerophospholipids, thereby creating 3′ phosphoinositides (PIs) [181]. PI3Ks consist of three subunits, namely, the p85 regulatory subunit, p55 regulatory subunit, and p110 catalytic subunit [182]. The family of PI3Ks can be divided into three classes, being class I, II, and III [183]. PI3Ks of class I can even be subdivided into class 1A and class 1B. Class 1A PI3Ks form heterodimers, containing the p58 and p110 subunits, being frequently implicated in different types of cancer, but they are also involved in regulating the development and maturation of the nervous system [184,185]. The p110 subunit can be expressed in three different isoforms: p110α, p110β, and p110δ catalytic subunits. However, the p110γ subunit is exclusively represented by class 1B PI3Ks. The p85 subunit encloses several variants such as p85a, p85b, and p85g, which are, respectively, encoded by the genes PIK3R1, PIK3R2, and PIK3R3 [186]. The different p85 subunit isoforms are almost all expressed in neurons. However, the p110α subunit is especially found in the hippocampus, olfactory bulb, and cerebellum [186,187]. Activation of PI3Ks is mediated by extracellular stimuli, such as growth factors, cytokines, and hormones, and are key mediators of biological processes, such as memory, neuronal survival, neurogenesis, and apoptosis [186,188]. Once fully active, PI3K phosphorylates PtdIns(4,5)P2 (PIP2) in order to generate the second messenger PtdIns(3,4,5)P3 (PIP3). PIP3 is able to bind other targets localized in the cell membrane via lipid–domain interactions, activating, e.g., the Akt-mTOR signaling pathway [66]. Counterbalancing the activation of the Akt-PI3K-mTOR pathway is mediated through dephosphorylation by protein phosphatase PTEN, thereby preventing downstream kinase signaling [189].

As mentioned above, PI3Ks are interconnected with signaling pathways involving Akt and mTOR. For this reason, PI3K activity is thought to be compromised in FXS. An early study by Hu et al. (2007) reported abnormalities in the Ras-PI3K-Akt signaling pathway in a mouse model for the FXS. The activity of PI3K was indirectly determined by western blotting of 3-phosphoinositide-dependent protein kinase-1 (PDK1), as it is a master kinase facilitating phosphorylation of Akt. Global expression of PDK1 was slightly increased in CA1 cells of Fmr1 knock-out mice compared to wild-types, however not significantly. Administration of histamine, a neuromodulator, increased phosphorylation of PDK1 at Thr308 in wild-type cultures, but failed to elevate phosphorylation of PDK1 in knock-out cultures, suggesting that the Ras-PI3K-Akt pathway is seriously impaired in the Fmr1 knock-out mouse [72]. A next study focused on elevated group 1 mGluR-mediated signaling and protein synthesis in the FXS and quantified PI3K, as it is a major player in downstream signaling of many cell surface receptors. Here, the predominant neuronal isoform of PI3K, being p110β, was immunoprecipitated from Fmr1 knock-out and wild-type cortical SNS. Next, the ability of p110β to catalyze the conversion of PI to PIP3 was investigated by radiolabeled ATP to quantify PI phosphorylation. The activity of PI3K was found to be increased in Fmr1 knock-out SNS compared to wild-type SNS. As an extra validation, the amount of PH-domain containing proteins was measured as accumulation of PIP2 and PIP3 leads to recruitment of lipid–domain interactions at synaptic membranes. Localization was enriched in Fmr1 knock-out but not in wild-type synapses, indicating overactivity of PI3K. To address the dependence of group 1 mGluRs on PI3K activity, similar experiments were performed in HEK293T cells, because they do not express group 1 mGluRs. The siRNA-mediated Fmr1 knockdown resulted in an increase in PI3K enzymatic activity together with elevated levels of Akt phosphorylation, suggesting that PI3K activity could also occur independently of group 1 mGluRs. In wild-type mice, activation of group 1 mGluRs resulted in p110β translation and expression together with an induced PI3K activity. Opposite findings were reported in the knock-out model, where protein synthesis and PI3K activity were increased and non-responsive to stimulation of group 1 mGluRs, suggesting abnormal PI3K signaling in the FXS. The PI3K antagonists LY294002 restored three key biochemical phenotypes associated with the FXS, being elevated synaptic protein synthesis, AMPA receptor internalization, and increased spine density. These results suggest targeting of PI3K as a potent strategy for FXS treatment [118]. After studying LTD in the Fmr1 knock-out mouse model, Sharma et al. (2010) reported that Akt phosphorylation was upregulated, which potentially could be achieved by its upstream kinase PI3K. Expression of PI3K was examined in the hippocampus of young Fmr1 knock-out and wild-type mice, revealing that the expression of the catalytic subunit was increased p110β in the Fmr1 knock-out model. The regulatory p85 subunit showed no difference in expression. Suppressing protein phosphatase PTEN was investigated, as it is known to inhibit PI3K activity. PTEN activity was assessed by western blotting for residues Ser380, Thr382, and Thr383 and was found to be decreased in the Fmr1 knock-out mouse. The higher expression of the p110β subunit and a decreased PTEN activity are both indicative of a higher PI3K activity in the Fmr1 knock-out model. Upstream of PI3K, the small nuclear GTPase PIKE is located, which is responsible for PI3K activation. PIKE is expressed as a nuclear isoform PIKE-S and a nucleocytoplasmic isoform PIKE-L. Expression of both isoforms were tested by western blotting in the Fmr1 knock-out hippocampus. The expression of the L-isoform was increased in the Fmr1 knock-out mice, while the expression of the S-isoform was unaffected. These results are in line with findings in wild-type mice, where FMRP is able to regulate PIKE-S expression. Hence, loss of FMRP would contribute to an elevation of PIKE-S levels, directly impacting PI3K activity [29]. Two years later, research performed by Gross was translated from the Fmr1 knock-out mouse model to human fragile X lymphoblastoid cell lines. They showed that protein synthesis is increased in patient cells using quantitative fluorescent metabolic labeling, similar to findings in the animal model. In addition, similar changes in expression and enzymatic activity of the PI3K catalytic subunit p110β could be observed in FXS patient cells and the Fmr1-deficient mouse model. Moreover, treatment with the p110β-selective antagonist TGX-221 resulted in a lower degree of phosphorylation of the PI3K catalytic subunit after western assessment in both murine and human wild-type and FXS SNSs. These findings of dysregulated expression and activity of the p110β catalytic subunit suggest the therapeutic potential for PI3K inhibitors in the clinic [190]. In contrast to previous research on PI3K activity, Lim et al. (2014) performed experiments with compounds that activate or inhibit the serotonin subtype 2B receptors, together with dopamine subtype 1-like receptors. Low dosage of these compounds activated the ERK-PI3K-Akt pathway, suggesting that PI3K activity is reduced in the Fmr1 knock-out mice. Treatment restored learning deficits without any anxiety-related side effects, suggesting that combined drug therapy could ameliorate the FXS phenotype [74]. In 2020, a new in silico approach was used, where public transcriptome databases were screened for the possible value for treatment of the Fmr1-deficient mouse model with the FDA-approved drug trifluoperazine. To address the increased levels of protein synthesis, newly synthesized proteins were labeled with puromycin. An increase in protein synthesis was observed in Fmr1 knock-out neurons. Administration of low-dose trifluoperazine rescued the aberrant translation in Fmr1 knock-out neurons and did not affect protein synthesis in wild-type neurons. The compound was compared for similarities in mechanism of action and revealed a close relationship with PI3K signaling inhibitors such as sirolimus and LY-294002. Hence, the effect of trifluoperazine treatment was assessed in Fmr1 knock-out and wild-type hippocampal neurons. A PI3K activity ELISA experiment showed a higher activity of PI3K in Fmr1-deficient cell cultures, which could be reduced by trifluoperazine treatment. Consistently, trifluoperazine also reduced the phosphorylation of two PI3K effectors, e.g., Ser473 of Akt and Thr389 of S6K1 in knock-out animals. These data indicate trifluoperazine as a potential treatment for FXS [191]. Recently, PI3K activity was studied in human-patient-induced pluripotent stem cell (iPSC)-derived neural progenitor cells and organoids as a model for FXS. Similarly to the previous study, elevated protein synthesis was observed in FXS-patient-derived cells compared to controls after protein tagging with the BONVAT or SUnSET method. To address whether PI3K activity is elevated during human neurogenesis in the FXS, expression of the catalytic subunit p110β was quantified by western blotting. Expression of p110β was significantly elevated in FXS patient-derived iPSCs as well as in the frontal cortex of FXS postmortem tissues compared to controls. Acute treatment with the p110β-specific inhibitor TGX-221 reduced protein synthesis in FXS-patient-derived iPSCs, indicative of a higher PI3K activity and abnormal signaling in FXS patient neural cells [192].

### 4.14. Protein Kinase cAMP-Activated Catalytic Subunit Alpha (PKA)

Protein Kinase cAMP-Activated Catalytic Subunit Alpha (**PKA**) is a highly conserved cyclic adenosine monophosphate (cAMP)-dependent kinase, which catalyzes the phosphorylation of serine, threonine, and tyrosine amino acids [193]. They function as key integrators of intracellular signaling and are essential in brain processes such as memory formation and cognition [194]. Structurally, PKA consists of two regulatory dimer-forming subunits, together with two bound catalytic subunits, forming an inactive tetrameric holoenzyme. To become fully active, PKA reacts to external stimuli that raise intracellular cAMP concentrations, which bind in a stoichiometry of two cAMP molecules to each regulatory subunit. The levels of cAMP increase after activation of adenylate cyclase upon GPCR activation. However, inhibition of cyclic nucleotide phosphodiesterase (PDE) enzyme will also increase cAMP levels. Binding of cAMP will induce a conformational change of the subunit dimer, releasing the catalytic subunit for downstream phosphorylation events [195]. PKA has been studied in multiple neurological diseases such as schizophrenia, bipolar disorders, and ASD [195,196,197].

The emphasis of PKA signaling in FXS has mainly focused on cAMP metabolism, and the protein kinase itself has been poorly studied in contrast to the other kinases mentioned above. In platelets derived from FXS patients, inconsistent results were found for cAMP accumulation. Some patients showed overlap with controls, while others showed reduced cAMP levels, reflecting the variability in phenotype of patients with the syndrome [198]. Later on, other researchers reported findings in line with the initial findings by Berry-Kravis and declared a robust defect in cAMP signaling in human, murine, or fly models. Hence, treatment with PDE-4 inhibitors seemed to have beneficial effects and could reverse some phenotypes associated with FXS [199,200,201,202]. Initial research of PKA in a Fmr1 knock-out mouse model was performed by Koga et al. (2015), investigating the role of FMRP in presynaptic long-term potentiation (LTP). The potentiation was shown to be blocked in the Fmr1-deficient mouse model after multichannel-electrode array recordings. Next, PKA subunits were investigated in three different protein fractions, being homogenate, cytosome, and synaptosome, of Fmr1 wild-type and knock-out mice. No changes in the expression of the catalytic subunit of PKA, nor the regulatory subunits RIIα and RIIβ, were found in the homogenate of knock-out mice compared to wild-types. The α-regulatory subunit also showed no difference in expression in the cytosome fraction. However, the catalytic PKA subunit was increased in the cytosolic fractions and was decreased in the synaptosomal fraction of Fmr1 knock-out mice. Moreover, the expression of the β-regulatory subunit was decreased in the synaptosome of knock-out mice, suggesting that loss of FMRP affects localization of PKA subunits across the synapse in the anterior cingulate cortex. In this study, researchers suggested that altered PKA subunit translocated could affect PKA activity in the Fmr1 knock-out mouse. However, actual activity studies were not performed [203]. In 2019, the activity of PKA was first studied in a *Drosophila* FXS disease model using an in vivo transgenic PKA activity reporter assay. Loss of FMRP resulted in a reduction in PKA activity in the central brain mushroom body (MB) circuit. However, FMRP overexpression positively correlated with elevated PKA activity. Moreover, loss of Rugose, a large A-Kinase Anchor Protein (AKAP) which can bind PKA, resulted in a reduction in PKA activity. As PKA signaling affects cytoskeleton dynamics, F-actin regulation was investigated. In the FXS fly model, F-actin was found to be accumulated in MB cells, and loss of Rugose also resulted in similar F-actin accumulation. These data suggest a FMRP-Rugose-PKA signaling mechanism that involves regulation of cytoskeleton dynamics, and proposes PKA as a new target for therapy in the FXS [204]. Findings by Sears et al. (2014) were extended in 2020 and showed that *Drosophila* and human FMRP enhance PKA activity in brain regions involved in memory and learning. Conversely, enhancing neuronal PKA activity negatively correlated with FMRP expression, demonstrating an FMRP-PKA negative feedback loop. Moreover, patient-derived R138Q point mutations of FMRP increased PKA activity in dendritic arbors, generated oxidative stress, and disrupted the neuronal cell architecture. The activation of PKA was also assessed by western blotting for its catalytic subunit, where phosphorylation of Thr197-198 was found to be increased, indicative of higher activity. Deletion of the RNA-binding RGG box of human FMRP (hFMRP) also reduced *Drosophila* FMRP (dFMRP), suggesting that FMRP does not exclusively regulate binding of its own mRNA in a direct negative self-regulation loop. The hFMRP-RGG mutant formed protein aggregates with colocalized PKA activity in neuronal cells. These data indicate that FMRP is a regulator of PKA activity and acts via a negative feedback loop regulating learning and memory [205]. Only recently, PKA activity was studied in the context of sex differences between male and female Fmr1 knock-out mice in response to L-stepholidine, a dopamine D1 receptor agonist and D2 receptor antagonist. Comparing phosphorylation of PKA in a sex-dependent way, female Fmr1 knock-out mice showed a strong reduction in phosphorylated PKA at Thr197, similarly to male knock-outs. Under physiological conditions, male wild-type mice showed a higher phosphorylation of PKA at Thr197 compared to wild-type females. However, Fmr1 knock-out animals show no sex differences in phosphorylated PKA. Daily injections of 10 mg/kg L-stepholidine elevated the phosphorylation of PKA in wild-type males, but not in female animals. In contrast, L-stepholidine treatment increased the phospho PKA status in female Fmr1 knock-out mice but decreased the phosphorylation in Fmr1 knock-out males. These findings suggest strong sex differences in Fmr1 knock-out mice, which translate to sex-associated differences in FXS patients, anticipating that future research of underlying kinase signaling in males and females could help to comprehend clinical benefits for all FXS patients [206].

### 4.15. Protein Kinase C (PKC)

Protein kinase C (**PKC**) represents a family of serine and threonine kinases that signal in a lipid-dependent manner, e.g., binding of phosphatidylserine. Currently, ten PKC isoforms have been identified and can be divided into three groups: (1) the conventional isoforms (α, β_I_, β_II,_ and γ), which become active after binding of second messengers Ca^2+^ and diacylglycerol (DAG), (2) novel isoforms (δ, ε, ζ, θ, and μ) that only depend on DAG for activation, and (3) atypical isoforms (ζ, ι, and λ) which are regulated by protein–protein interactions independent of Ca^2+^ and DAG [207,208]. Structurally, PKCs contain a regulatory subunit and a catalytic subunit, residing at the C-terminus which ensures kinase activity. Here, a highly conserved ATP and magnesium binding site are present, as well as a binding domain for phosphorylated substrate proteins. At the N-terminus, the regulatory subunit is present, which has two different biochemical conformational states. In an inactive state, the regulatory subunit, consisting of the C1 and C2 domains, is bound to the catalytic region, blocking its enzymatic activity. Activation of PKC results in a conformational change, relieving the inhibitory interaction, which subsequently ensures PKC activity. Over the years, aberrant PKC activity has been reported in cancer, diabetes, heart failure, pain, and neurological disorders, such as Alzheimer’s disease and bipolar disorder [209].

Recently, changes in PKC signaling have also been reported in FXS. Treatment with the GABA_B_ receptor agonist Baclofen was shown to be effective for improving symptoms in an Fmr1-deficient mouse model and human FXS patients. However, the signaling mechanisms underlying the GABA_B_ receptor and FMRP still need further investigation. Zhang et al. (2015) reported that activation of the GABA_B_ receptor results in Fmrp expression by upregulation of cAMP response element binding protein (CREB) in primary mouse cultures by two mechanisms: (1) transactivation of insulin-like growth factor-1 receptor (IGF-1R), and (2) activation of PKC. To assess PKC activity, phosphorylation of the downstream substrate MARCKS was investigated by western blotting. Phosphorylation of MARCKS at Ser152/156 was increased in a time-dependent manner upon baclofen stimulation but was reduced after treatment with the FAK inhibitor PF573228, suggesting that PKC acts downstream of FAK. Next, the effects of three different PKC inhibitors were tested on CREB activation. All three were capable of reducing CREB phosphorylation, indicating that PKC is required for CREB activation after GABA_B_ receptor stimulation. Next, the effect of GABA_B_ receptor activation was studied for induction of FMRP expression. Upon GABA_B_ receptor stimulation with baclofen, FMRP synthesis was decreased. Similar findings were reported for IGF-1R, suggesting that both IGF-1R and PKC are crucial players in the expression of Fmrp after GABA_B_ receptor activation. Unfortunately, there are no reports of PKC activity studies in Fmr1-deficient cultures [210]. In 2015, another study investigated the GABA_A_ receptor subunits in vitro and in vivo in Fmr1 knock-out and wild-type mice. It found a significant decrease in the expression of α1 GABA_A_ receptor and PKC in Fmr1 knock-out mice compared to wild-type mice. Moreover, phosphorylation of α1 GABA_A_ receptor at Ser408/409 and PKC at Thr497 were both reduced in cortical neurons and brains from Fmr1 knock-out mice, indicative of a lower activity. To address whether α1 GABA_A_ receptor is mediated by PKC, a specific inhibitor calphostin C was used and showed a threefold decrease in α1 GABA_A_ receptor phosphorylation. The effect of the PKC activator PDBu was also tested in cortical cultures and showed a time-dependent increase in phosphorylation of the α1 GABA_A_ receptor. Thus, studies with a PKC inhibitor as well as an activator suggest that phosphorylation of the α1 GABA_A_ receptor is mediated by PKC [211]. Another method for determining kinase activity is performing electrophysiological experiments. Deng and Klyachko (2016) demonstrated that the pyramidal cells in the entorhinal cortex of Fmr1 knock-out mice show an aberrant action potential threshold and increased excitability. The alternations in threshold are accounted by an increased persistent sodium current (I_NaP_), which is mediated by an increased mGluR5-PLC-PKC signaling. Therefore, PKC activity was investigated for the influence of the I_NaP_ using two potent and selective PKC inhibitors, calphostin C and PKC19–36. Using a recording pipette, administration of both calphostin C and PKC19–36 reduced the excess I_NaP_ current of the Fmr1 knock-out mouse back to baseline wild-type levels. These findings indicate I_NaP_ current deregulation in the FXS and hyperexcitability, which can be targeted with specific PKC inhibitors [212]. In 2020, a study by Geoffroy et al. investigated loss of FMRP and its association with DGKk-PKC signaling, leading to an excess of the second messenger DAG in the cortical neurons of an FXS mouse model and in the cerebellum of an FXS patient. A direct effector of DAG is PKC, and hence the activity of PKC was determined using western blotting. An increase of PKC phosphorylation at Ser660 was measured in cortical neurons of Fmr1 knock-out mice compared to wild-type cultures. Administration of pioglitazone (PGZ), a molecule reported to inhibit the DAG–PKC signaling pathway by activating DGK activity, corrected the elevated phosphorylation in knock-out cultures, while wild-type cultures remained unaffected. Moreover, PGZ treatment ameliorated behavioral alternations associated with the FXS phenotype. These findings suggest dysregulation of lipid homeostasis and support PGZ treatment for improving symptoms in FXS patients [104].

### 4.16. p90 Ribosomal Protein S6 Kinase (RSK)

The p90 ribosomal protein S6 kinases (**RSKs**) are a serine-threonine kinase family that play a role in various cellular processes such as cell growth, motility and migration, survival, and proliferation, thereby functioning as downstream mediators of the MAPK signaling cascade [213]. Currently, the kinase family comprises four members of RSK1-4 and two related homologues RSK-like protein kinases. The four RSK isoforms are activated by ERK1/2 signaling in response to external stimuli such as hormones and neurotransmitters [214]. In its inactive conformation, RSKs reside in the cytoplasm and nucleus together with their upstream activator ERK1/2. However, activation of ERK1/2 results in phosphorylation and activation of RSKs in the plasma membrane, after which it accumulates in the nucleus [215]. The structure of RSKs is highly conserved among the isoforms and they are all composed of an N-terminal kinase domain and a C-terminal kinase domain, interconnected by a linker region. Activation of RSKs by ERK1/2 results in increased phosphorylation of several tyrosine and serine amino acid residues (Ser221, Thr359, Ser363, Ser380, Thr573, and Ser749). The C-terminal contains a D-domain, enabling docking of ERK1/2 [216,217]. Phosphorylation of Ser221 occurs in the activation loop at the N-terminal kinase domain and is mediated by PDK1 [218]. Amino acid residues Ser363 and Ser380 are localized in the linker region, where Ser363 is phosphorylated by ERK1/2, and undergoes autophosphorylation or plausible phosphorylation mediated by heterologous kinases. Phosphorylation of motif Ser380 is thought to be exerted by the C-terminal kinase domain [215]. The Thr573 phosphorylation site resides in the activation loop of the C-terminal kinase domain and is activated by ERK1/2, contributing to RSK translocation to the plasma membrane [219]. The RSK family is especially interesting in the brain, since loss-of-function mutations of the RSK2 gene result in Coffin–Lowry syndrome, a form of intellectual disability characterized by key features such as psychomotor deficits, facial dysmorphisms, skeletal malformations, and cognitive impairment [220].

Sawicka et al. (2016) reported higher activity of ERK-induced p90 ribosomal S6 kinase activity in Fmr1 knock-out mice. Here, defective ERK signaling was studied outside of the hippocampus, where ERK appears normal with no change in basal activity [119,190]. Subsequently, ERK-mTOR-S6 kinase signaling was examined in the neocortex of an Fmr1 knock-out mouse model. Here, ERK signaling was found to be elevated in the cortex of Fmr1 knock-out mice with a higher phosphorylation at Thr185/Tyr187, indicative for a higher activation, while mTOR phosphorylation remained unchanged at Ser2448. Next, S6 kinase phosphorylation was assessed in Fmr1 knock-out mice at the age of four weeks, where higher phosphorylation was reported at Ser235/236, an RSK- and S6K-dependent site. However, phosphorylation of S6 at Ser240/244, an RSK-independent but S6K-dependent site, remained unchanged, suggesting that S6K is not the responsible kinase for S6 phosphorylation. Since S6 is stronger phosphorylated on a RSK-dependent site, an alternative signaling mechanism was hypothesized. Hence, phosphorylation of RSK was measured in order to see whether it is stronger activated in Fmr1 knock-out mice. RSK expression was unchanged in knock-out versus wild-type mice. However, phosphorylation of RSK at the ERK1/2-dependent site Thr359/Ser363 as well as the autophosphorylation site Ser380 were increased in the neocortex of Fmr1 knock-out mice compared to controls. Next, the ERK1/2-RSK signaling was studied in cortical SNS of Fmr1 knock-out mice. Here, phosphorylation of Thr185/Tyr187 of ERK1/2, pSer380 of RSK, and Ser235/236 of S6 were increased in Fmr1 knock-out mice. No changes in protein expression of either ERK, RSK, and S6 were reported in cortical SNS of wild-type and knock-out mice. Apart from S6 phosphorylation, the RSK family is also capable of phosphorylating other protein substrates such as the Shank family, which are involved in postsynaptic scaffolding and synaptic transmission both strongly associated with ASD [221]. Surprisingly, both Shank1 and Shank3 contain an RSK consensus phosphorylation site, suggesting that RSK can induce Shank phosphorylation. Immunoprecipitation with an antibody against the RSK consensus site and western blotting with an antibody that recognizes all phosphorylation sites of Shank resulted in a higher phosphorylation of Shank in cortical synaptosomes of Fmr1 knock-out mice, indicating that RSK activity is associated with Shank phosphorylation in the FXS. In order to see whether the higher RSK signaling persisted with the onset of age, the activity was determined again at the adult age of 10 weeks. Here, phosphorylation of MEK, ERK, MNK, eIF4E, and RSK was increased in Fmr1 knock-out mice, indicating a persistent higher activity of RSK. In order to counterbalance the elevated RSK activity, Fmr1 wild-type and knock-out mice were treated with the blood–brain permeable RSK inhibitor BI-D1870. Acute administration of the RSK inhibitor prevented audiogenic seizures in Fmr1 knock-out mice, indicating that an increased RSK activity contributes to the FXS phenotype and forms a target for therapy [77].

### 4.17. p70 Ribosomal Protein S6 Kinases (S6Ks)

The p70 ribosomal protein S6 kinases (**S6Ks**) are part of a conserved serine-threonine kinase family that mainly function as downstream effectors of the mTOR pathway [222]. S6Ks become active upon activation of mTOR in response to nutrients, growth factors, stress, and cellular energy, thereby stimulating protein synthesis and cell proliferation [223]. Defects in S6K signaling can contribute to pathological conditions such as cancer, obesity, diabetes, and neurodevelopmental disorders, including down syndrome, tuberous sclerosis, ASD, and Rett syndrome [224,225]. The S6K protein family consists of two members, S6K1 and S6K2, that share a high homology in their kinase domain [224]. Structurally, both isoforms consist of an N-terminal domain containing an NLS, a kinase domain with an Thr229 activation site, activated by PDK1, a linker region with three phospho-amino acid residues (Ser371, Thr389, and Thr404), which can be phosphorylated by mTORC1, together with a C-terminal domain, which can be phosphorylated at the residues Ser411, Ser418, Ser421, and Ser424 [223,226]. Functionally, the C-terminal domain interacts with the N-terminal domain, forming an autoinhibitory loop preventing kinase signaling [227]. Activation of S6K occurs with phosphorylation of the four C-terminal serine residues, thereby exposing the linker region which is subsequently accessible for mTORC1 phosphorylation. Ultimately, phosphorylation mediated by PDK1 is performed at the threonine residue of the kinase domain, activating S6K [223,224,228,229]. Both S6K isoforms are implicated in diverse biological processes such as cytoskeleton reorganization, protein synthesis, signal transduction, transcription, and splicing, which are all dysregulated in the FXS [223,230].

As S6K is a direct effector of mTOR signaling, Hoeffer et al. (2012) investigated altered signaling mechanisms in lymphocytes from FXS patients and healthy subjects. First, phosphorylation of S6K1 was assessed by western blotting, resulting in a stronger Thr389 phosphorylation quantified to total S6K1 expression in FXS lymphocytes compared to controls. Next, a direct S6K substrate, the ribosomal protein S6, was investigated, resulting in a higher phosphorylation at Ser 235/236 in FXS lymphocytes with no difference in expression of total S6. Controversially, after investigating mTOR, phosphorylation of Ser2448, a site that also includes S6K action, remained unaffected between patients and controls. To determine whether the higher phosphorylation of S6K was able to induce translation, eIF4E phosphorylation was investigated, as it is a substrate of S6K and participates in translation initiation. Here, a massive increase in phosphorylation of eIF4E at Ser209 was found in FXS patients, while total eIF4E expression was not affected. These findings were further explored in post-mortem brain tissue of FXS patients and healthy age-matched controls. Similarly to findings in lymphocytes, S6K1 phosphorylation was increased at Thr389 in brains of FXS patients. No alterations in phosphorylation of mTOR were observed. Phosphorylation of effector substrate eIF4E at Ser209 was extremely low in FXS patients, possibly explained by high phosphatase activity in the brain during the post-mortem interval. Total eIF4E abundance was unchanged between patients and controls and was easy to detect. These results indicate abnormal translation in lymphocytes and post-mortem brain tissue from FXS patients [121]. Concurrently, research performed by Bhattacharya et al. (2012) investigated the enhanced translation in a mouse model for the fragile X syndrome. Genetic removal of S6K1 by crossing an Fmr1 knock-out mouse with an S6k1 knock-out mouse reduced the enhanced phosphorylation of proteins involved in translation back to wild-type levels. For example, S6 was phosphorylated more strongly in Fmr1 knock-out mice at Ser240/244 and Ser235/236 compared to wild-types. Genetic ablation of S6K1 lowered the degree of S6 phosphorylation back to baseline. Phosphorylation of eIF4E was also increased in Fmr1 knock-out mice, which could be corrected in the Fmr1/S6k1 double knock-out mouse. As S6K1 is also known to act on mTOR and ERK1/2 signaling pathways, phosphorylation of these protein kinases was assessed in all genotypes. Phosphorylation of mTOR was increased in Fmr1 knock-out mice compared to wild-types, and genetic reduction of S6K1 rescued the increased phosphorylation. Similarly, ERK1/2 phosphorylation was increased in Fmr1 knock-out mice which could be restored by genetic reduction of S6K1. Moreover, mGluR LTD, as well as body weight and macroorchidisms, were corrected. Removal of S6K1 prevented deficits in social interaction, novel object recognition, and autism-associated behavior. These results indicate S6K1 kinase activity as a key player in the pathophysiology of the FXS [120]. Additional research on S6K was performed after studying increased cerebral protein synthesis in Fmr1 knock-out mice, where dietary lithium was used to improve mood alterations and correct the elevated protein synthesis. Lithium treatment had no effect on phosphorylated S6K1 nor on total S6K1 abundance in the hippocampus of Fmr1 wild-type and knock-out mice. Basal phosphorylation of S6K1 and total S6K1 expression were also unaltered without chronic lithium treatment, indicating that lithium treatment had little effect on the elevated cerebral protein synthesis [73]. SK6 signaling was also shown to be regulated by the alternative protein kinase p90 ribosomal S6 kinase (RSK) after investigation of audiogenic seizure susceptibility in Fmr1 knock-out mice. Here, phosphorylation, and thus activation, of S6 was enhanced for the RSK-dependent, respectively, the S6K-dependent site Ser235/236 in the neocortex of four-week-old Fmr1 knock-out mice. However, phosphorylation at the S6K-dependent, but RSK-independent sites Ser240/244 showed no significant difference between Fmr1 knock-out and wild-type mice. To discriminate which upstream kinase was responsible for the elevated S6 phosphorylation, the phosphorylation status of both S6K and RSK were determined. First, S6K phosphorylation at Thr389 was unchanged in Fmr1 knock-out mice and control littermates, suggesting that S6K is not the main upstream kinase accounting for hyperphosphorylation of S6. Secondly, the alternative ERK-mediated kinase RSK was investigated in cortical SNS of Fmr1-deficient and wild-type mice. Here, phosphorylation at Ser380 was significantly increased relative to total RSK abundance in Fmr1 knock-out mice. Furthermore, phosphorylation of S6 was increased at Ser235/236, but not at the S6K-dependent Ser240/244 in Fmr1 knock-out mice, indicating that elevated RSK activity at cortical synapses results in higher protein phosphorylation. Pharmacological inhibition of ERK corrected the elevated RSK signaling and S6 activity, revealing a multimodal aspect of S6 in the cortical synapse of Fmr1 knock-out mice, where it integrates signals of multiple upstream kinases. Inhibition of RSK with the specific blood–brain permeable inhibitor BI-D1870 showed a reduction in audiogenic seizures in Fmr1 knock-out mice, thereby identifying RSK as a potential target for treatment [77]. Later, research was performed on the mTORC1 pathway, where mice carrying the Fmr1 deletion were chronically treated with rapamycin, an mTORC1 inhibitor. Here, lysates of the frontal cortex of vehicle- and rapamycin-treated Fmr1 knock-out mice were analyzed by western blotting and analyzed for the effect on genotype and treatment on phosphorylated targets relative to total protein abundance. No effect of rapamycin treatment was observed for either genotype or treatment on phospho p70S6K. Next, two phosphorylation sites of S6 were examined, where both phosphorylation of Ser235/236 and Ser240/244 were significantly elevated for genotype and treatment interaction. Subsequently, a post hoc analysis revealed that phospho-S6 was enhanced in Fmr1 knock-out mice compared to vehicle-treated controls. Rapamycin treatment was only able to reduce phosphorylated levels of S6 in Fmr1 knock-out mice for the Ser235/236 sites with no effect on the controls, suggesting that rapamycin treatment could ameliorate the behavioral phenotype of Fmr1 knock-out mice. However, rapamycin treatment had an adverse effect on sleep and social behavior in both genotypes, concluding that targeting the mTORC1 pathway, and hence phosphorylation of S6, seems not to be the best treatment strategy [169]. Recently, an in silico transcriptome profiling of Fmr1-deficient mouse neurons highlighted the potential of the drug trifluoperazine (TFP) and revealed that TFP could normalize the elevated protein synthesis, which is a key defect in the FXS. Furthermore, TFP was able to inhibit the PI3K-Akt-S6K signaling cascade, forming the pathway affecting the protein translation machinery which was shown to be hyperactivated in Fmr1 knock-out mice and FXS patients. Treatment with TFP for one hour reduced S6K1 phosphorylation at the PI3K-target site Thr389 in wild-type neuronal cultures. Earlier studies often reported on the elevated S6K activity, and these findings were also validated in Fmr1 knock-out cultures in this study, where they showed that Fmr1-deficient hippocampal cultures exhibited an increased S6K1 phosphorylation. Administration of TFP reduced the enhanced phosphorylation back to wild-type levels, demonstrating that S6K1 signaling is enhanced in Fmr1 knock-out hippocampal cultures, and also suggests the antipsychotic drug TFP as potential treatment for the FXS [191].

### 4.18. Tropomyosin-Related Kinase B (TrkB)

The tropomyosin-related kinases (**Trks**) are a family of receptor protein tyrosine kinases (RTKs) consisting of the members tropomyosin-related kinase A (TrkA), B (TrkB), and C (TrkC) [231]. The family of Trks shows high affinity for several neurotrophins, which are crucial growth factors involved in neuronal survival and development. Examples include neurotrophins such as brain-derived neurotrophic factor (BDNF), NT-4, nerve growth factor (NGF), and neurotrohphin-3 (NT-3), which each show preference for a specific Trk receptor family member, e.g., BDNF is the primary ligand of TrkB, and to a lesser extent NT-4 [232]. The TrkB signaling is initiated after neurotrophin binding (e.g., BDNF), causing tyrosine receptor dimerization and autophosphorylation of the tyrosine residues of the intracellular kinase domain [233]. Activation of the internal kinase domain evokes phosphorylation of tyrosine residues on the C-terminus of the receptor, which subsequently can act as a docking site for small adaptor molecules. For example, phosphorylation of Tyr515 causes recruitment of Shc adaptor molecules through their phosphotyrosine binding domains (PTB) [234]. Tyr816 phosphorylation causes binding of PLCγ through the adaptor molecule Src-homology 2 (SH2). Several other proteins containing pleckstrin homology (PH) and SH2 domains can also function as adaptor molecules that bind phosphorylated tyrosine residues, and mediate TrkB receptor activation [235]. Recruitment of these adaptor molecules causes binding of growth factor receptor bound protein 2 (GRB2) and/or son of sevenless (SOS) which mainly activates (1) the MAPK pathway, (2) the PI3K–Akt pathway, and (3) the PLCγ–Ca^2+^ pathway, ultimately resulting in transcription of genes involved in neuronal differentiation, proliferation, synaptic plasticity, growth, and cell survival [236,237]. Expression of TrkB can be found in the central (CNS) as well as in the peripheral nervous system (PNS). Regions in the CNS that contain high TrkB expression include the cerebral cortex, hippocampus, frontal cortex, visual system, plexus choroideus, brain stem, and spinal cord. On the other hand, the dorsal root ganglia, cranial ganglia, and the vestibular systems are regions in the PNS that show high TrkB expression. Non-neuronal tissues such as liver, kidney, pancreas, and skeletal muscles also show TrkB expression [231,237,238].

Initially, FMRP was discovered to be regulated by the neurotrophin BDNF. In vitro administration of BDNF to cultured rat hippocampal neurons for 24 h resulted in a downregulation of FMR1. Moreover, transgenic mice which overexpressed the active TrkB receptor showed a slight downregulation of Fmr1 mRNA in the hippocampus. However, expression of BDNF and TrkB remained unchanged in Fmr1 knock-out and control mice. Activity of the TrkB receptor was not assessed in Fmr1-deficient mice [239]. Later, immunohistological analyses of GABA-releasing interneurons showed that TrkBs were predominantly found in dendritic extensions of wild-type sections, while Fmr1 knockout sections revealed a significant reduction in the amount of TrkBs. In contrast, parvalbumin (PV) interneurons of Fmr1 knock-out mice showed an upregulation of TrkB in the soma compared to wild-type controls, indicating specific changes in the GABAergic interneuron circuit [240]. BDNF/TrkB actions were also assessed in undifferentiated neuronal progenitor cells (NPCs) of Fmr1-deficient mice. Here, changes in differentiation and migration of TrkB expressing neurosphere cultures, as well as in the developing cortex, were observed together with abnormal intracellular responses to calcium and ATP. Expression of BDNF was reduced in the cortex but enhanced in the hippocampus of Fmr1 knock-out mice. However, functional activity studies implicating the TrkB receptor or its phosphorylation status were not investigated [241]. Other research further indicates a decrease in BDNF-TrkB signaling after investigating maturation of fast-spiking interneurons in Fmr1 knock-out mice. Here, expression of BDNF was reduced in the cortex of five-day-old Fmr1 knock-out pups, but the TrkB receptor was upregulated in knock-outs compared to wild-type controls, suggesting a deficit in signaling during the critical period of development. Next, BDNF-TrkB signaling was investigated by intraperitoneal injection of the TrkB-agonist LM22A, which induced increased phosphorylation at Tyr515 of TrkB in both genotypes. The increased phosphorylation was normalized with coadministration of the TrkB antagonist ANA12, which was also validated with electrophysiological experiments, suggesting reduced neurotrophin signaling during early development in Fmr1 mice [242]. In another study, pharmacological inhibition with istradefylline, an adenosine A_2A_ receptor (A_2A_R) antagonist, revealed a reduction in synaptic and cognitive abnormalities in Fmr1 knock-out mice. Similarly to a behavioral rescue, overactive signaling pathways such as the mTOR, TrkB, and STEP signaling were corrected. Here, the truncated form of TrkB was found to be increased in the hippocampus of Fmr1 knock-out mice, which could be normalized by A_2A_R antagonization. TrkB receptor truncation could inhibit downstream tyrosine kinase signaling, thereby affecting multiple downstream signaling processes. Ultimately, reduced BDNF levels were found in the hippocampus of Fmr1 knock-out mice, and reduction of the truncated-TrkB receptor partially rescued the Fmr1 mouse phenotype [243].

## 5. A Predictive Network Analysis Suggests Functional Associations between the Abnormally Regulated Protein Kinases in the FXS

Mapping of kinase interaction networks is essential for understanding the cellular processes that protein kinases participate in, and subsequently could provide information about disease-associated signaling pathways [244]. For this reason, the UniProt IDs of the protein kinases whose activity was reported in FXS literature were submitted to the STRING database version 11.5 to identify network interactions (Figure 4). Several functional associations of FMRP and all the above-mentioned protein kinases were characterized by protein homology, co-expression, text mining, and database evidence. Generally, functional enrichment analysis revealed that the submitted kinases were strongly connected (protein–protein-interaction (PPI) enrichment *p* < 0.0001). Here, FMRP is involved in a tightly related network of protein kinases with its main interactors being LIMK1, CaMKIIα, Akt1, S6K1, GSK3β, and TrkB. For DGKk, no functional associations with other protein kinases or with FMRP were predicted.

## 6. The Aberrant Kinase-Signature in the Fragile X Syndrome Translates to Abnormalities of the Phosphoproteome

Intracellular signaling is mostly mediated through phosphorylation of protein kinases, which are dysregulated in the FXS. However, despite great advances in studies of protein kinases in the FXS, an understanding of their dysfunction is far from clear and needs an integrative approach. For this reason, identification of phosphorylated proteins can be exerted by mass spectrometry-based phosphoproteomics. In this way, the activity of protein kinases can be indirectly quantified with phosphoproteomics as changes in protein phosphorylation result from dynamic alterations in protein kinase signaling [245]. Initially, quantitative proteome analyses have been performed in mouse and *Drosophila* models for the FXS, which were developed to identify differences in synaptic protein expression. Stable isotope labeling by amino acids in cell culture (SILAC)-based proteomics in *Fmr1* knock-out and wild-type cortical neurons resulted in identification of 132 differentially expressed proteins in absence of FMRP, related to changes in synaptic structures, neurotransmission, and dendritic mRNA transport, together with autism- and epilepsy-related proteins [246]. Another study compared the proteomes of *Fmr1* knock-out and wild-type hippocampal synapses, using isobaric tags for relative and absolute quantitation (iTRAQ). A series of 23 proteins were significantly different in expression with groups of proteins known to be involved in cellular differentiation, neurite outgrowth, and synaptic vesicle release [247]. Another SILAC study performed in a heterozygous *dfmr1 Drosophila* model resulted in profiling of 1617 proteins, identifying several proteins which were altered in expression such as actin-binding protein profilin and microtubule-associated protein futsch [248]. A final study investigated systemic protein expression in neocortical synaptic fractions from *Fmr1* knock-out and wild-type mice at adolescent and adult stages of life. Over 100 proteins were upregulated in *Fmr1*-deficient mice at adolescence, but this was no longer the case in adult mice. The differentially expressed proteins were involved in processes affecting brain development, ASD, and intellectual disability, which were further integrated in an interactome showing a central role for PSD95 [249]. Besides these proteomic attempts to resolve global molecular processes in FXS, phosphoproteomic studies are still poorly applied in FXS, although phosphorylation abnormalities might be critical determinants of FXS pathologies, based on the above-mentioned kinase literature. Only one proteome-wide study on the phosphorylation abnormalities was reported, in which SILAC-based quantitative phosphoproteomics was used to analyze murine *Fmr1* knock-out and wild-type fibroblastic cell lines derived from *Fmr1*-deficient embryos to identify proteins and phosphorylation sites dysregulated as a consequence of FMRP loss. FMRP-related changes in the levels of 5023 proteins and events were initially identified and mapped onto major signaling transduction pathways. This study confirmed the global downregulation of the MAPK/ERK pathway and a decrease in the phosphorylation level of ERK1/2 in the absence of FMRP, which is connected to attenuation of long-term potentiation. Additionally, this study revealed phosphorylation abnormalities in several additional pathways, as it detected differential expression and phosphorylation of proteins involved in pre-mRNA processing and nuclear transport, as well as Wnt and calcium homeostasis signaling, which involves signaling by PLC, PKC, NFAT, and cPLA2 [250].

## 7. Concluding Remarks and Future Directions

In this review, we summarized the current knowledge on protein kinase signaling, phosphorylation abnormalities, and phosphoproteomics-based approaches in the FXS, highlighting the complexity of its pathophysiology. We demonstrate the interplay of post-translational modifications (PTMs) involved in the regulation of FMRP, necessary for strict translational control in healthy neurons. However, further investigations are required to determine the effects of other PTMs besides phosphorylation, and whether these PTMs also result in dramatic phenotypic effects in the absence of FMRP. Subsequently, we have extensively discussed the protein kinases and phosphatases involved in the FXS. This summary highlights the complexity of capturing dynamic changes in kinase signaling networks which sometimes may yield conflicting results at the single kinase level. On top of this, many studies on protein kinases have been performed on different neurodevelopmental stages of life, model organisms, sample material, and techniques, which challenges data integration to better understand the mechanisms governing the FXS. We also explored a gap in the FXS literature, being the lack of integrative phosphoproteomics studies, which could provide novel molecular insights into the etiology of the disease. Besides phosphoproteomics approaches, integrative kinase activity assays such as NanoBret^TM^ (Promega, Madison, WI, USA), Pamchip (Pamgene, ‘s Hertogenbosch, The Netherlands) or multiplexing phosphorylation assays (FullMoon Biosystems, Sunnyvale, CA, USA) may provide novel perspectives on kinase network signaling defects which contribute to the FXS physiopathology. In addition, several of the identified kinases in the *Fmr1* knock-out mouse model can be integrated with pathway analysis tools to find new targets for kinase inhibitor treatment. Additionally, translational studies are required to extrapolate the findings from the *Fmr1* knock-out mouse model to different FXS patient sample materials. However, the capacity of FMRP to bind directly to different kinase mRNAs, e.g., DGKκ, could contribute to a potential feedback regulation, limiting kinase inhibitor therapy. Currently, specific kinase inhibitors such as a selective PI3K subunit inhibitor or different GSK3 isoform inhibitors have already been implemented for treatment with great success in the *Fmr1* knock-out mouse model. Nevertheless, clinical trials in FXS patients tend to fail, possibly due to a lack of a proper outcome measure, indicating a high demand for objective assessment of FXS clinical trials. Thus, designing a generalized and patient-specific kinase signature could provide insights into the molecular signaling networks of the syndrome and could even help to stratify these within the wide heterogeneous group of FXS patients. One possible way to approach this would be to test different FXS patient-derived cell lines in response to a selection of kinase inhibitors on a phosphorylation array. In this way, an accurately informed inhibitor response profile can be created for controls and FXS patients, providing a first step towards personalized medicine. Furthermore, stratifications of patients, as well as defining kinase inhibitor responder profiles associated with different patient backgrounds, remains necessary for the development of personalized medicines in FXS. Additional research is required to obtain a global kinome tree activity map to facilitate future implementation of kinase inhibitors and kinase activity-based biomarkers for personalized FXS treatment.

## Figures and Tables

**Figure 1 cells-11-01325-f001:**
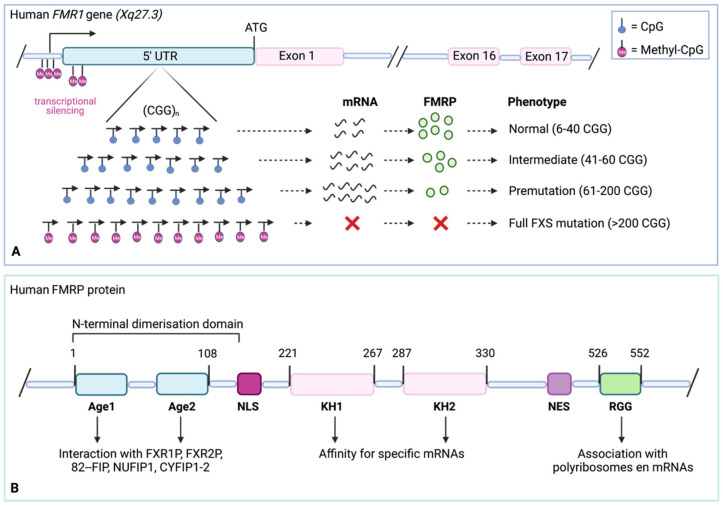
FMR1 gene structure and FMRP protein domains. (**A**) The human *FMR1* gene located on chromosome Xq27.3. *FMR1* is defined by seventeen exons with different methylation states of CpG islet located in the 5′UTR, resulting in distinct Fragile X-related phenotypes. The 5′UTR is the common site of an expansion mutation in the FXS. (**B**) Human FMRP contains several conserved protein domains such as K homology domains (KH1, and 2 together with an RGG box which enable mRNA binding capacity. The Agenet (Age) domains act on histones and trimethylated lysine residues and facilitate FMRP protein–protein interactions.

**Figure 2 cells-11-01325-f002:**
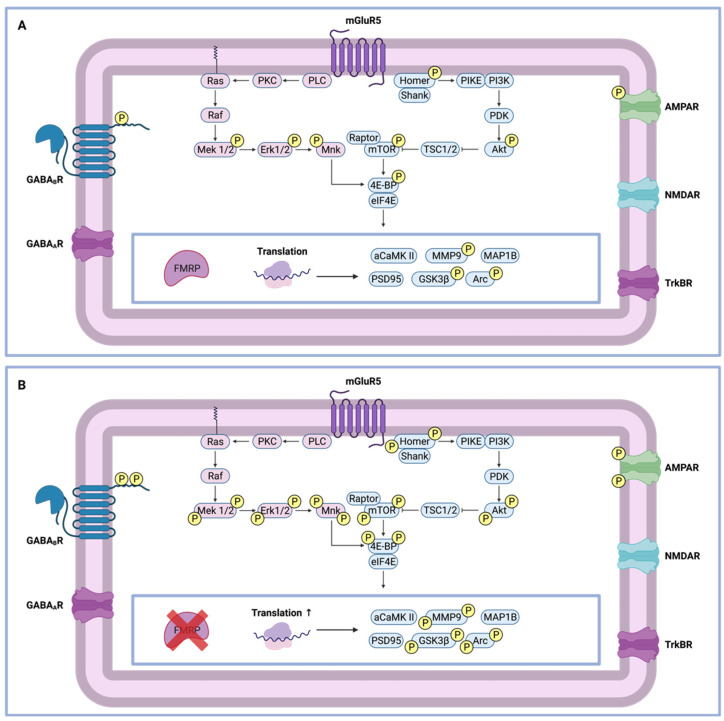
Schematic overview of the molecular signaling pathways in the FXS. (**A**) In physiological conditions, the glutamatergic pathway is activated upon binding of glutamate to the mGluR5 which is located on the postsynaptic cell membrane. A network of protein kinases and phosphatases, with their downstream signaling targets, either becomes activated or inhibited due to phosphorylation, thus regulating the activity of FMRP. Phosphorylation of FMRP results in the release of the bound mRNAs, causing protein translation. (**B**) Loss of FMRP causes a broad-range kinomic deregulation which results in a stronger phosphorylation of several protein kinase substrates, e.g., the GABAergic pathway is affected by loss of FMRP, because it stabilizes mRNAs of several subunits of the GABA_A_ receptor. Loss of FMRP also causes translational imbalance with an increased synthesis of proteins such as Calcium/Calmodulin Dependent Protein Kinase II Alpha (αCaMKII), Matrix Metallopeptidase 9 (MMP9), Microtubule Associated Protein 1B (MAP1B), postsynaptic density protein 95 (PSD95), Glycogen Synthase Kinase 3 Beta (GSKβ), and Activity Regulated Cytoskeleton Associated Protein (Arc).

**Figure 3 cells-11-01325-f003:**
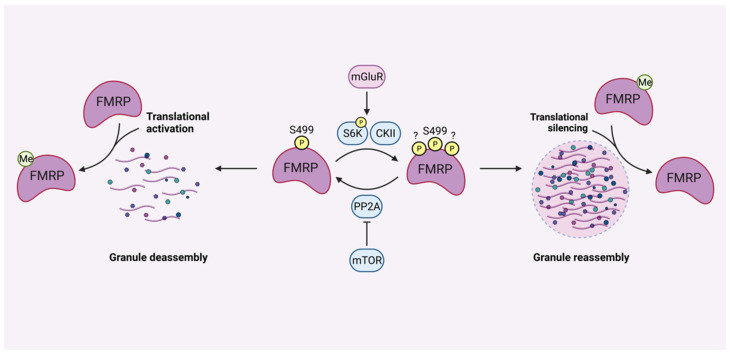
Kinase-phosphatase regulation of the Fragile X Messenger Ribonucleoprotein. FMRP is regulated by a kinomic network and exists in two biochemical states, being phosphorylated by ribosomal protein S6 kinase 1 (S6K1) or dephosphorylated by protein phosphatase 2A (PP2A). Other post-translational modifications, such as methylation, ensure neuronal granule disassembly and translational activation. However, S6K1 phosphorylation promotes FMRP demethylation and facilitates granule formation by high-complexity protein interactions.

**Figure 4 cells-11-01325-f004:**
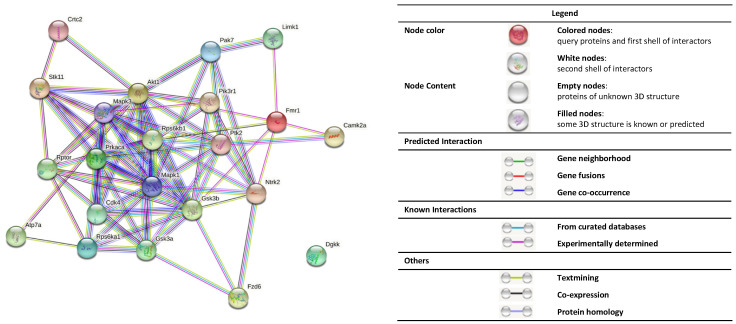
Network interaction of protein kinases that are dysregulated by loss of FMRP. A protein interaction network was generated by String Version 11.5 and summarizes the predicted associations of all the protein kinases mentioned in the literature regarding FXS. Protein kinases are represented by colored network nodes in relation to each other and to the Fragile X-causing protein FMRP. The edges illustrate functional associations and the lines between the nodes represent existence of evidence for associations. A more detailed legend of the node colors, content, and predicted and known interactions is provided in the graphic legend.

**Table 1 cells-11-01325-t001:** Overview of the FXS studies examining dysregulated protein kinases.

Protein Kinase	Observed Activity	Experimental Specifications	Study Reference
Protein kinase B (**PKB**/**Akt**)	No difference	*Fmr1* knock-out mouse hippocampal extracts from cultures and brain	(Hu et al., 2008)
	Increased	*Fmr1* knock-out mouse hippocampus	(Sharma et al., 2010)
	Increased	*Fmr1* knock-out mouse hippocampus	(Liu et al., 2012)
	Increased	*Fmr1* knock-out mouse hippocampal cultures	(Lim et al., 2014)
	Increased	*Fmr1* knock-out mouse hippocampal neuron cultures	(Sidhu et al., 2014)
	Increased	Human FXS platelets	(Pellerin et al., 2016)
	No difference	*Fmr1* knock-out mouse neocortex	(Sawicka et al., 2016)
	Increased	*Fmr1* knock-out mouse primary hippocampal cultures	(Ding et al., 2020)
AMP-activated protein kinase (**AMPK**)	Increased	Insulin-producing cells (IPCs) of the brain of a dFMR1 *Drosophila* model	(Monyak et al., 2016)
	No difference	Prefrontal cortex and hippocampus of *Fmr1* knock-out mouse	(Gantois et al., 2017)
	Not reported	*Fmr1* knock-out mouse primary hippocampal cultures	(Yan et al., 2018)
Calcium/calmodulin-dependent protein kinase type II subunit alpha (**CaMKIIα**)	Increased	*Fmr1* knock-out mouse primary hippocampal and neocortex cultures	(Guo et al., 2015)
Cyclin-dependent kinase 4 (**CDK4**)	Not reported	*Fmr1* knock-out adult neural progenitor/stem cell cultures	(Luo et al., 2010)
Diacylglycerol Kinase Kappa (**DGKk**)	Decreased	*Fmr1* knock-out mouse primary cortical neuron cultures	(Tabet et al., 2016)
	Decreased	FXS patient cerebellum postmortem	(Tabet et al., 2016)
	Decreased	*Fmr1* knock-out mouse primary cortical neuron cultures	(Geoffroy et al., 2020)
	Decreased	*Fmr1* knock-out mouse primary cortical neuron cultures	(Habbas et al., 2021)
Extracellular signal-regulated kinases (**ERK1/2**)	Increased	*Fmr1* knock-out mouse hippocampus	(Hou et al., 2006)
	No difference	*Fmr1* knock-out mouse spinal cord (ERK2)	(Price et al., 2007)
	No difference	*Fmr1* knock-out mouse hippocampal extract from cultures and brain	(Hu et al., 2008)
	Decreased	*Fmr1* knock-out mouse cortical synaptoneurosome	(Kim et al., 2008)
	No difference	*Fmr1* knock-out mouse cortical synaptoneurosome	(Gross et al., 2010)
	Increased	*Fmr1* knock-out mouse hippocampus	(Osterweil et al., 2010)
	Increased	*Fmr1* knock-out mouse hippocampus	(Bhattacharya et al., 2012)
	No difference	*FXS* lymphocytes	(Hoeffer et al., 2012)
	Increased	*Fmr1* knock-out mouse adult brain	(Rubin et al., 2014)
	Increased	Human FXS platelets	(Pellerin et al., 2016)
	Increased	*Fmr1* knock-out mouse neocortex	(Sawicka et al., 2016)
	Increased	*Fmr1* knock-out mouse primary hippocampal cultures	(Ding et al., 2020)
Focal adhesion kinase (**FAK**)	Increased	*Fmr1* knock-out mouse hippocampal neuron cultures	(Sidhu et al., 2014)
Glycogen synthase kinase 3 (**GSK3**)	Increased	GSKβ in *Fmr1* knock-out mouse striatum and cortex	(Min et al., 2009)
	Increased	GSKα in *Fmr1* knock-out mouse hippocampus, striatum and cortex	(Min et al., 2009)
	Increased	*Fmr1* knock-out mouse brain	(Mines et al., 2010)
	Increased	*Fmr1* knock-out mouse hippocampus	(Yuskaitis et al., 2010)
	Increased	*Fmr1* knock-out mouse hippocampus	(Choi et al., 2016)
	Increased	*Fmr1* knock-out mouse cortex	(McCamphill et al., 2020)
LIM Domain Kinase 1 (**LIMK1**)	Increased	*dFMR1 Drosophila*	(Kashima et al., 2016)
	Increased	Human postmortem FXS patient prefrontal cortex	(Kashima et al., 2016)
	Increased	*Fmr1* knock-out mouse somatosensory cortex primary cultures	(Pyronneau et al., 2017)
MAPK Interacting Serine/Threonine Kinase 1 (**MNK**)	Increased	*Fmr1* knock-out mouse hippocampus and cortex	(Shukla et al., 2021)
	Increased	*Fmr1* knock-out mouse hippocampal slices	(Gkogkas et al., 2014)
	Increased	*Fmr1* knock-out mouse hippocampus and cortex	(Shukla et al., 2021)
Mechanistic Target of Rapamycin Kinase (**mTOR**)	No difference	*Fmr1* knock-out mouse hippocampus	(Osterweil et al., 2010)
	No difference	*Fmr1* knock-out mouse cortex	(Sharma et al., 2010)
	Increased	*Fmr1* knock-out mouse hippocampus	(Sharma et al., 2010)
	Increased	*Fmr1* knock-out mouse hippocampus	(Bhattacharya et al., 2012)
	No difference	*FXS* lymphocytes	(Hoeffer et al., 2012)
	Increased	*Fmr1* knock-out mouse hippocampus	(Liu et al., 2012)
	Increased	*Fmr1* knock-out mouse hippocampus	(Busquets et al., 2013)
	Increased	*Fmr1* knock-out mouse primary hippocampal cultures	(Sidhu et al., 2014)
	Increased	*Fmr1* knock-out mouse hippocampus	(Choi et al., 2016)
	No difference	*Fmr1* knock-out mouse neocortex	(Sawicka et al., 2016)
	No difference	*Fmr1* knock-out mouse frontal cortex	(Saré et al., 2018)
	Increased	*Fmr1* knock-out mouse primary hippocampal cultures	(Yan et al., 2018)
p21 (RAC1) Activated Kinases (**PAKs**)	Increased	*Fmr1* knock-out mouse forebrain	(Hayashi et al., 2007)
	Increased	*Fmr1* knock-out mouse forebrain	(Dolan et al., 2013)
Phosphatidylinositol-4,5-Bisphosphate 3-Kinase (**PI3K**)	Increased	*Fmr1* knock-out mouse hippocampal extracts from cultures and brain	(Hu et al., 2008)
	Increased	*Fmr1* knock-out mouse cortical synaptoneurosomes	(Gross et al., 2010)
	Increased	*Fmr1* knock-out mouse hippocampus	(Sharma et al., 2010)
	Increased	Human *FXS* lymphoblastoid cell cultures	(Gross and Bassell, 2012)
	Decreased	*Fmr1* knock-out mouse hippocampal cultures	(Lim et al., 2014)
	Increased	*Fmr1* knock-out mouse primary hippocampal cultures	(Ding et al., 2020)
	Increased	Human iPSC derived neural *FXS* model	(Raj et al., 2021)
Protein Kinase cAMP-Activated Catalytic Subunit Alpha (**PKA**)	No reported	*Fmr1* knock-out mouse anterior cingulate cortex	(Koga et al., 2015)
	Decreased	*dfmr1* null MB Kenyon cells	(Sears et al., 2019)
	Increased	*dfmr1* null MB Kenyon cells	(Sears et al., 2020)
	Increased	Male *Fmr1* knock-out mice	(Jiang et al., 2021)
Protein Kinase C (**PKC**)	No reported	*Fmr1* knock-out mouse cultured cerebellar granule neurons	(Zhang et al., 2015)
	Decreased	*Fmr1* knock-out mouse cortical neurons and brains	(Zhao et al., 2015)
	Increased	*Fmr1* knock-out mouse entorhinal cortex	(Deng and Klyachko, 2016)
	Increased	*Fmr1* knock-out mouse primary cortical neuron cultures	(Geoffroy et al., 2020)
p90 Ribosomal Protein S6 Kinase (**RSK**)	Increased	*Fmr1* knock-out mouse neocortex	(Sawicka et al., 2016)
p70 Ribosomal Protein S6 Kinase (**S6K**)	Increased	*FXS* lymphocytes	(Hoeffer et al., 2012)
	Increased	*Fmr1* knock-out mouse hippocampus	(Bhattacharya et al., 2012)
	No difference	*Fmr1* knock-out mouse hippocampus	(Liu et al., 2012)
	Increased	*Fmr1* knock-out mouse neocortex	(Sawicka et al., 2016)
	No difference	*Fmr1* knock-out mouse frontal cortex	(Saré et al., 2018)
	Increased	*Fmr1* knock-out mouse primary hippocampal cultures	(Ding et al., 2020)
Tropomyosin-related kinase B (**TrkB**)	Not reported	*Fmr1* knock-out mouse primary hippocampal cultures	(Castrén et al., 2002)
	Not reported	*Fmr1* knock-out mouse somatosensory cortex	(Selby et al., 2007)
	Not reported	*Fmr1* knock-out mouse cortical neuronal progenitor cells	(Louhivuori et al., 2010)
	Not reported	*Fmr1* knock-out mouse neocortex	(Louhivuori et al., 2010)
	Decreased	*Fmr1* knock-out mouse cortex	(Nomura et al., 2017)
	Decreased	*Fmr1* knock-out mouse hippocampus	(Ferrante et al., 2021)

## Data Availability

Not applicable.

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
