# Peer review of "Towards Kinase Inhibitor Therapies for Fragile X Syndrome: Tweaking Twists in the Autism Spectrum Kinase Signaling Network"

_cells, 2022, doi:10.3390/cells11081325_

Round 1

Reviewer 1 Report

This is a review of how phosphorylation and dephosphorylation regulates the FMRP protein network and regulates function in the context of Fragile X disease.

This is a well written and very detailed review, maybe too detailed and long.  As general comments:

1) Maybe figures are not sufficiently self-explanatory and are somewhat difficult to follow.  Either the figure should be improved or the figure legend be more detailed.

2) Very useful table with FXS effect on protein kinases within the FMRP network.

3) Maybe consider shortening the manuscript to make it easier to read and with important points of each molecule highlighted.

Author Response

Point 1: Maybe figures are not sufficiently self-explanatory and are somewhat difficult to follow. Either the figure should be improved, or the figure legend be more detailed.

Response 1: We have improved the figures and even simplified figures that were too complex. The legend was written in more detail, for example figure 2.

Point 2: Very useful table with FXS effect on protein kinases within the FMRP network.

Response 2: Thank you.

Point 3: Maybe consider shortening the manuscript to make it easier to read and with important points of each molecule highlighted.

Response 3: We have improved the quality of reading by excluding sentences that were repeated amongst different kinases, for example overlap between Akt, ERK and mTOR. However, we do not want to shorten the manuscript, especially because we want to highlight the complexity of kinase research in FXS.

Reviewer 2 Report

This article reviews the breadth of protein kinase and phosphatase studies performed in FXS. It contains a detailed description of the current knowledge of cell signaling in fragile X models while highlighting the complexity of the pathophysiology. Table 1 is particularly helpful in summarizing the data. The paper concludes by identifying a lack of data regarding integrative phosphoproteomics. The article is well written and thorough. I have a few minor suggestions for improvement:

(1) The title states, Toward kinase inhibitor therapies for Fragile X syndrome…, and the conclusion mentions personalized treatment. In between there is an immense amount of data regarding protein kinases and phosphatases. The article would greatly benefit by discussion of how the current data, and the proposed acquisition of more data, will be integrated to choose the best therapeutic target(s) for personalized medicine for FXS considering the overlap between the signaling pathways and feedback loops.

(2) The manuscript needs to be thoroughly checked for typos and grammatical errors. Some are listed below.

(3) There seems to be duplication of some the discussion, for example S6K, lithium, and mTORC information.

Line 75-76 Are citations 19 and 20 mixed up?

Line 116 The stated repeat sizes disagree with Figure 1

Line 149 Could comment on recent findings FMRP acts as an m6A reader protein to shuttle mRNA targets between nucleus and cytoplasm

Line 232 What is meant by membraneless vesicles?

Line 387 FMRP binding to kinase mRNAs adds another layer of potential feedback regulation that could be discussed in conjunction with treating with inhibitors and potential impact on personalized medicine at the end of the paper

Line 422 spelling signaling

Line 438 spelling phosphorylation

Line 441 Is ASS supposed to be ASD?

Line 445 capitalize C-terminal

Line 469, 473 spelling DHPG

Line 486 spelling assessed

Line 454, 490, 517, 580, 764, 796, 803, 948, 976, 984, 1054, 1078, 1086, 1471 can delete the word “the” before FXS

Line 507 spelling converging

Line 508 spelling solely

Line 522 spelling regulated

Line 523 grammar more strongly

Line 528 spelling phase

Line 557, 699, 703 , 730, 792, 813, 846, 851, 857, 863, 873, 945, 1006, 1024, 1100, 1242, 1346, 1350, 1449 inconsistent nomenclature for Fmr1 knock-out – sometimes use fragile X mice

Line 567-568 spelling reside

Line 576 spelling lose

Line 660 There is an additional relevant citation: PMCID: PMC8529056

Line 697, 987, 1030, 1253, 1264 inconsistent use of the acronym for synaptoneurosome

Line 743 spelling fine tunes

Line 748 spelling exerts

Line 755 spelling proteins

Line 778 spelling independently

Line 843 spelling Fmr1 should be italicized

Line 846 spelling disturbed

Line 888 can use FXS acronym

Line 893 are fragile X neurons, referring to human cells or Fmr1KO mouse neurons?

Line 909 spelling up until now

Line 932 spelling autophosphorylated

Line 936 spelling crucial

Line 938 spelling inability

Line 998, 1021 first use of Fmr1 wild type mice, previously using wild type mice

Line 1017 spelling nucleocytoplasmic

Line 1023 spelling later

Line 1037 spelling anxiety

Line 1066 spelling function

Line 1067 spelling in

Line 1087 capitalize X

Line 1090 spelling fractions

Line 1081 spelling for

Line 1083 spelling later on

Line 1098 spelling FMRP

Line 1119 first use of hFMRP and dFMRP, acronyms not spelled out

Line 1124 Fmr1 not italicized

Lines 1171, 1173, 1460, 1463, 1465, 1468 FMRP or Fmrp

Line 1197 DGKk acronym spelled out earlier

Line 1280 capitalize Rett

Table 1 line 4 Fmr1 not italicized

The acronym GPCR is spelled out multiple times.

Citations: some only have initials and not surnames of the authors

Author Response

Point 1: The title states, Toward kinase inhibitor therapies for Fragile X syndrome…, and the conclusion mentions personalized treatment. In between there is an immense amount of data regarding protein kinases and phosphatases. The article would greatly benefit by discussion of how the current data, and the proposed acquisition of more data, will be integrated to choose the best therapeutic target(s) for personalized medicine for FXS considering the overlap between the signaling pathways and feedback loops.

Response 1: Thank you for this objective feedback. We have included this important remark in the discussion. Briefly, we suggest for example that one should test different FXS patient-derived cell lines in response to a selection of kinase inhibitors on a phosphorylation array. In this way, an accurately informed inhibitor response profile can be created for controls and FXS patients, providing a first step towards personalized medicine.

Point 2: The manuscript needs to be thoroughly checked for typos and grammatical errors. Some are listed below.

Response 2: Our excuses. We thoroughly checked for typos and grammatical errors, and even approach our internal medical writing devision as an extra check on top of ourselves.

Point 3: There seems to be duplication of some the discussion, for example S6K, lithium, and mTORC information.

Response 3: We are aware that this is a problem. Together with our internal medical writing devision we tried to overcome this overlap partially. However, it is also necessary to mention part of the discussion for some kinases, because the outcome on target phosphorylation can be completely different.

Point 4: Line 75-76 Are citations 19 and 20 mixed up?

Response 4: Citations were mixed up and re-adjusted. 

Point 5: Line 116 The stated repeat sizes disagree with Figure 1

Response 5: We adjusted the figure according to the written text.  

Point 6: Line 149 Could comment on recent findings FMRP acts as an m6A reader protein to shuttle mRNA targets between nucleus and cytoplasm.

Response 6: we have included and discussed the following papers

  1. Zhang et al., “Fragile X mental retardation protein modulates the stability of its m6A-marked messenger RNA targets,” Hum. Mol. Genet., vol. 27, no. 22, pp. 3936–3950, Nov. 2018, doi: 10.1093/HMG/DDY292.
  2. M. Edens et al., “FMRP Modulates Neural Differentiation through m 6 A-Dependent mRNA Nuclear Export,” Cell Rep., vol. 28, no. 4, pp. 845-854.e5, Jul. 2019, doi: 10.1016/J.CELREP.2019.06.072.

Point 7: Line 232 What is meant by membraneless vesicles?

Response 7: We have altered these words by ribonucleoprotein organelles.

Point 8: Line 387 FMRP binding to kinase mRNAs adds another layer of potential feedback regulation that could be discussed in conjunction with treating with inhibitors and potential impact on personalized medicine at the end of the paper

Response 8: Good remark. We have introduced this comment in the discussion: for example, translational studies are required to extrapolate the findings from the Fmr1 knock-out mouse model to different FXS patient sample materials. However, the capacity of FMRP to bind directly to different kinase mRNAs e.g., DGKκ, could contribute to a potential feedback regulation, limiting kinase inhibitor therapy.

Point 9: Line 422 spelling signaling/Line 438 spelling phosphorylation/Line 441 Is ASS supposed to be ASD? Line 445 capitalize C-terminal/Line 469, 473 spelling DHPG/Line 486 spelling assessed/Line 454, 490, 517, 580, 764, 796, 803, 948, 976, 984, 1054, 1078, 1086, 1471 can delete the word “the” before FXS/Line 507 spelling converging/Line 508 spelling solely/Line 522 spelling regulated/Line 523 grammar more strongly/Line 528 spelling phase/Line 557, 699, 703 , 730, 792, 813, 846, 851, 857, 863, 873, 945, 1006, 1024, 1100, 1242, 1346, 1350, 1449 inconsistent nomenclature for Fmr1 knock-out – sometimes use fragile X mice/Line 567-568 spelling reside/Line 576 spelling lose/Line 660 There is an additional relevant citation: PMCID: PMC8529056/Line 697, 987, 1030, 1253, 1264 inconsistent use of the acronym for synaptoneurosome/Line 743 spelling fine tunes/Line 748 spelling exerts/Line 755 spelling proteins/Line 778 spelling independently/Line 843 spelling Fmr1 should be italicized/Line 846 spelling disturbed/Line 888 can use FXS acronym/Line 893 are fragile X neurons, referring to human cells or Fmr1KO mouse neurons?/Line 909 spelling up until now/Line 932 spelling autophosphorylated/Line 936 spelling crucial/Line 938 spelling inability/Line 998, 1021 first use of Fmr1 wild type mice, previously using wild type mice/Line 1017 spelling nucleocytoplasmic/Line 1023 spelling later/Line 1037 spelling anxiety/Line 1066 spelling function/Line 1067 spelling in/Line 1087 capitalize X/Line 1090 spelling fractions/Line 1081 spelling for/Line 1083 spelling later on/Line 1098 spelling FMRP/Line 1119 first use of hFMRP and dFMRP, acronyms not spelled out/Line 1124 Fmr1 not italicized/Lines 1171, 1173, 1460, 1463, 1465, 1468 FMRP or Fmrp/Line 1197 DGKk acronym spelled out earlier/Line 1280 capitalize Rett/Table 1 line 4 Fmr1 not italicized/The acronym GPCR is spelled out multiple times.

Response 9: All these remarks, errors and adjustments were incorporated in the manuscript.

Point 10: Citations: some only have initials and not surnames of the authors.

Response 10: We corrected this uniformely.